# Benchmarking LLM-Assisted Blue Teaming via Standardized Threat Hunting

Yuqiao Meng [1]   Luoxi Tang [1]   Feiyang Yu [2]   Xi Li [3]   Guanhua Yan [1]   Ping Yang [1]   Zhaohan Xi [1]

## Abstract

As cyber threats continue to grow in scale and sophistication, blue team defenders increasingly require advanced tools to proactively detect and mitigate risks. Large Language Models (LLMs) offer promising capabilities for enhancing threat analysis. However, their effectiveness in real-world blue team threat-hunting scenarios remains insufficiently explored. This paper presents CYBERTEAM, a benchmark designed to guide LLMs in blue teaming practice. CYBERTEAM constructs a standardized workflow in two stages. First, it models realistic threat-hunting workflows by capturing the dependencies among analytical tasks from threat attribution to incident response. Next, each task is addressed through a set of operational modules tailored to its specific analytical requirements. This transforms threat hunting into a structured sequence of reasoning steps, with each step grounded in a discrete operation and ordered according to task-specific dependencies. Guided by this framework, LLMs are directed to perform threat-hunting tasks through modularized steps. Overall, CYBERTEAM integrates 30 tasks and 9 operational modules to guide LLMs through standardized threat analysis. We evaluate both leading LLMs and state-of-the-art cybersecurity agents, comparing CYBERTEAM against open-ended reasoning strategies. Our results highlight the improvements enabled by standardized design, while also revealing the limitations of open-ended reasoning in real-world threat hunting.

## 1. Introduction

The increasing frequency and sophistication of cyber threats continue to pose significant challenges to organizational security. In 2024 alone, over 11,000 more (38% increase!) vulnerabilities were reported compared to 2023, as evidenced by the MITRE CVE database (The MITRE Corporation, n.d.). Defenders, commonly known as the **blue team** (Diogenes & Ozkaya, 2018; Rajendran et al., 2011), are under increasing pressure to identify, analyze, and respond to malicious activities in a timely and accurate manner, a process termed **threat hunting**.

Recent advances in large language models (LLMs) have demonstrated impressive potential to augment cybersecurity practices, including malware analysis (Abusitta et al., 2021; Qian et al., 2025; Devadiga et al., 2023), penetration testing (Deng et al., 2024; Happe & Cito, 2023), and fuzzing (Zhang et al., 2025; Oliinyk et al., 2024; Black et al., 2024). However, those works tend to focus on isolated analytical tasks.

On the contrary, threat hunting is inherently multi-staged given the chained nature of cyber threat progression. Rather than a single point of analysis, practical threat hunting consist of a sequence of dependent tasks from understanding threat patterns, behaviors to incident response (mitigation), wherein the output of one hunting phase serves as the input context for the next. A blue team analyst cannot effectively contain or eradicate a threat without first accurately reconstructing the adversary's path through the network. Hence, the fragmented focus in current LLM research limits our understanding of how these models perform within complex, interdependent threat-hunting workflows.

To address this gap, we introduce CYBERTEAM, a practical benchmark designed to rigorously evaluate and guide the use of LLMs in blue team threat hunting. CYBERTEAM supports blue team efforts through the following aspects:

**Broader Coverage.** CYBERTEAM is constructed from a diverse and large-scale repository of threat intelligence data sourced from 23 vulnerability databases, including MITRE (MITRE Corporation, 2024), NVD (National Institute of Standards and Technology (NIST), 2024), and CISE (CISE Program, 2024), as well as security platforms such as Red Hat Bugzilla (Red Hat, Inc., 2024), Oracle Security Alerts (Oracle Corporation, 2024), and IBM X-Force (IBM Corporation, 2024). In addition, CYBERTEAM presents a larger

[1]State University of New York at Binghamton, Binghamton, NY, USA [2]Duke University, NC, USA [3]University of Alabama at Birmingham, AL, USA. Correspondence to: Zhaohan Xi <zxi1@binghamton.edu>.

*Proceedings of the 43rd International Conference on Machine Learning*, Seoul, South Korea. PMLR 306, 2026. Copyright 2026 by the author(s).

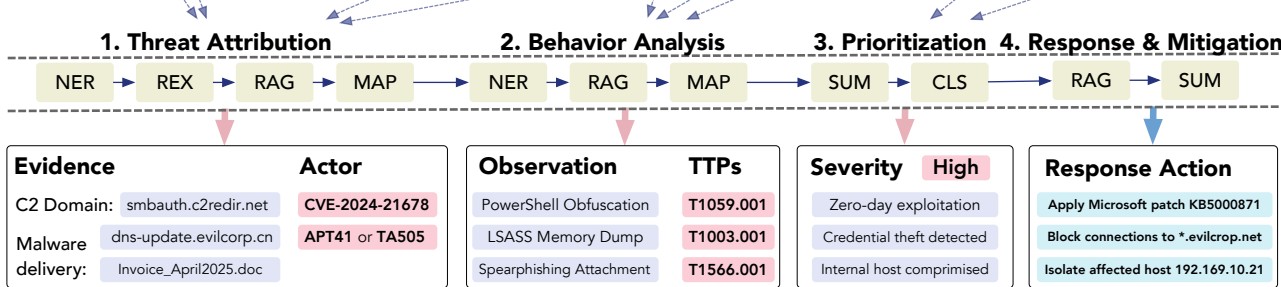

**Cyber Threat Log**

On Dec. 10, 2024, our SIEM system flagged multiple anomalous outbound DNS requests from internal host host-192-168-10-21.local to dns-update.evilcorp.net. Investigation revealed that the host had received a suspicious email containing an attachment named Invoice_April2025.doc, which, when opened, triggered a connection to a known C2 domain via an obfuscated PowerShell script. The initial vector appears to be a phishing campaign exploiting. The attacker leveraged PowerShell to execute a memory-resident payload that conducted system reconnaissance, credential harvesting (via LSASS dump), and lateral movement using SMB.
**Detected IOCs include: C2 Domains:** dns-update.evilcorp.cn, smbauth.c2redir.net. **IP Addresses:** 185.100.87.21, 192.168.10.22

*Figure 1.* A CYBERTEAM threat hunting example equipped with operational modules. Module names: NER–named entity recognition, REX–regex parsing, MAP–text mapping, RAG–retrieval-augmented generation, CLS–classification, SUM–summarization.

*Table 1.* Comparison between CYBERTEAM and other LLM-oriented cybersecurity benchmarks.

| Benchmark | Focus | #Data | #Task | #Source | Coverage | Unique Feature |
|---|---|---|---|---|---|---|
| CWE-Bench-Java (Li et al., 2025) | Java vulnerability | 120 | 4 | 1 | Four CWE classes | Large-scale Java codes |
| CTIBench (Alam et al., 2024) | Cyber Threat Intelligence | 2,500 | 3 | 6 | CVE, CWE, CVSS, ATT&CK | Multi-choice questions (MCQ) |
| SevenLLM-Bench (Ji et al., 2024) | Report understanding | 91,401 | 28 | N/A | Bilingual instruction corpus | Synthetic Data, MCQ, QA |
| SWE-Bench (Jimenez et al., 2024) | Software bug fixing | 2,294 | 12 | 1 | GitHub issues | Python repository |
| **CYBERTEAM (Ours)** | Blue team threat hunting | 452,293 | 30 | 23 | Threat-hunting lifecycle (3.1) | Open Generation, Standardized Reasoning Env |

number of tasks and samples than existing cybersecurity benchmarks (Jimenez et al., 2024; Li et al., 2025; Alam et al., 2024; Ji et al., 2024), as summarized in Table 1. This extensive coverage allows for a more comprehensive and nuanced evaluation of LLM performance across a wide range of threat-hunting scenarios.

**Standardized Workflow.** An important feature of CY-BERTEAM is its structured, modular workflow for guiding LLMs within a standardized reasoning environment (Yang et al., 2024). **This design is inspired by blue team practices, where analysts typically follow standardized procedures to interpret threat metadata and conduct investigations** (Sehgal & Thymianis, 2023; Diogenes & Ozkaya, 2018; Brotherston et al., 2024). However, strict adherence to such procedures can limit flexibility when analyzing unstructured threat logs or addressing emerging, zero-day threats. **To balance standardization and flexibility**, CYBERTEAM integrates a set of operational modules that regulate LLM behavior while allowing for open-ended reasoning where needed. As illustrated in Figure 1, CY-BERTEAM first models the dependency structure among threat-hunting objectives (e.g., attribution, behavior analysis, mitigation) as a task chain, and then maps this chain into a corresponding sequence of operational modules. In this process, functions such as NER enforce structured outputs (e.g., extracting threat actor entities), while functions

like RAG support more flexible reasoning (e.g., summarizing relevant patching strategies).

**Evaluation Strategy.** CYBERTEAM incorporates agent-based evaluation strategies tailored to each threat-hunting objective. We benchmark leading LLMs and state-of-the-art (SOTA) cybersecurity agents, comparing CYBERTEAM 's modularized approach with popular open-ended reasoning strategies such as In-Context Learning (ICL) (Dong et al., 2024), Chain-of-Thought (CoT) (Wei et al., 2022), Tree-of-Thought (ToT) (Yao et al., 2023). Our evaluation provides insights into the actionable threat hunting across 30 tasks.

In summary, this paper makes the following contributions: (1) We introduce CYBERTEAM, a practice-informed, broadly scoped benchmark that enables rigorous evaluation of LLMs for blue team threat hunting, (2) we construct a standardized reasoning workflow that models the dependencies among threat-hunting tasks and guides LLMs through standardized yet flexible reasoning workflow, (3) we conduct comprehensive evaluations and provide insights to improve LLM performance among threat-hunting scenarios.

## 2. Related Work

**LLMs for Cybersecurity.** Recently, LLMs have shown promise in enhancing cybersecurity tasks such as malware

classification (Abusitta et al., 2021; Qian et al., 2025; Devadiga et al., 2023), code vulnerability detection (Russell et al., 2018; Lu et al., 2024; Sheng et al., 2024), penetration testing (Happe & Cito, 2023; Shen et al., 2025), phishing detection (Kulkarni et al., 2025; Greco et al., 2024), and incident report generation (Bernardi et al., 2024; Sufi, 2024; McGregor et al., 2025). These applications leverage the language understanding and reasoning capabilities of LLMs to analyze technical data, recommend solutions, or simulate attacker behaviors. However, existing applications typically target isolated tasks without considering broader analyst workflows. Additionally, their open-ended reasoning often results in hallucinations and inconsistencies (Mündler et al., 2024), raising concerns about reliability in high-stakes defensive scenarios.

**Cybersecurity Benchmarks.** Recent benchmarks have focused on static analysis (Reinhold et al., 2024; Higuera et al., 2020; Braga et al., 2017), software vulnerabilities (Hasanov et al., 2024), and threat report generation (Tihanyi et al., 2024; Perrina et al., 2023; Čupka et al., 2023). These benchmarks evaluate predefined tasks such as identifying CWE categories, matching CVEs, or summarizing intelligence reports (Alam et al., 2024; Aghaei et al., 2020; Branescu et al., 2024; Hemberg et al., 2024). While helpful for reproducibility, they often cover narrow domains and lack the complexity and task interdependencies inherent in real-world threat investigations. In contrast, benchmarks from other high-stakes fields (e.g., law, medicine, finance) increasingly include complex, multistep tasks requiring diverse reasoning skills (Fei et al., 2024; Wang et al., 2024; Choshen et al., 2024; Lucas et al., 2024). Inspired by these efforts, we introduce CYBERTEAM to emphasize structured reasoning and realistic interdependencies for blue teaming scenarios.

**Operation-Guided Agents.** Recent research has proposed agents with operational modules to structure LLM reasoning into modular, interpretable steps (Driess et al., 2023; Dongre et al., 2025; Hu et al., 2025). Such frameworks have achieved notable success in robotics (Jeong et al., 2024; Akkaladevi et al., 2021), database querying (Kadir et al., 2024), and scientific reasoning tasks (Abate et al., 2020; Vaesen & Houkes, 2021). However, their use in cybersecurity, especially defensive operations, remains underexplored despite the need for structured workflows. Our work addresses this gap by introducing a modular environment aligned with blue team practices, enabling procedural reasoning within a structured analytical pipeline.

## 3. CYBERTEAM

In this section, we provide a detailed introduction of CYBERTEAM regarding the collected threat hunting tasks (3.1), data sources (3.2), and the modularized strategy (3.3).

### 3.1. Threat Hunting Tasks

As shown in Table 2, CYBERTEAM reflects the full life-cycle of threat hunting tasks. Specifically, CYBERTEAM systematizes analytical tasks into four categories: **Threat Attribution**, **Behavior Analysis**, **Prioritization**, and **Response & Mitigation**. Each category captures a stage in the threat-hunting workflow from investigating cyber threats to identifying countermeasures. Specifically:

**Threat Attribution** aims at uncovering the origins and nature of a threat. This includes tasks such as extracting infrastructure artifacts (e.g., domains, IPs, URLs), classifying malware families based on observed behaviors, matching known threat signatures, and linking activities to known campaigns or actor groups (e.g., APT or MITRE ATT&CK (MITRE Corporation, 2024)). Further granularity is achieved through geographic and temporal pattern analysis, as well as victimology and affiliation linking, all of which help analysts contextualize incidents in terms of their broader threat landscape.

Subsequently, **Behavior Analysis** focuses on understanding how adversaries interact with systems over time. Tasks in this category include mapping unusual file system activities, profiling network behaviors (e.g., Monitoring outbound traffic), detecting credential access, and analyzing the use of commands and scripts. Analysts aim to reconstruct sequences of attack events and associate them with specific execution contexts or behavioral patterns.

When multiple threats emerge simultaneously, **Prioritization** assesses their relative urgency and associated risk. This involves analyzing the attack vector and complexity, identifying privilege requirements and user interaction dependencies, and estimating potential impact. These factors are then synthesized into impact labels and severity scores (e.g., CVSS (FIRST, a)) to guide effective triage. Finally, **Response & Mitigation** focus on generating actionable defense strategies. This includes recommending response playbooks, generating patch code, correlating relevant security advisories, and suggesting appropriate tools or configuration changes to neutralize the threat.

### 3.2. Data Sources

CYBERTEAM collects threat metadata from two primary sources: (1) vulnerability databases, which offer authoritative structural and non-structural information about threats, and (2) threat intelligence platforms, which report event-driven, context-rich threat data.

**Vulnerability databases** serve as foundational resources for automated threat hunting by providing machine-readable records of software flaws, exposure types, and critical con-

*Table 2.* Threat hunting tasks, description of targets, corresponding operations, number of instances, and evaluation metrics. Details of implemented operations and involved metrics are in Appendix B and C, respectively.

| Task | Analytical Target | Standard Operation | #Data | Metric |
|---|---|---|---|---|
| *Threat Attribution* | | | | |
| Malware Identification | Malware delivery or toolset | NER, SUM + Reasoning | 15,742 | F1 |
| Signature Matching | Techniques from known threat groups | NER, SIM + Reasoning | 5,166 | F1 |
| Temporal Pattern Matching | Known work schedules | REX + Reasoning | 4,203 | Sim |
| Affiliation Linking | Source organizations | NER, MAP + Reasoning | 17,583 | F1 |
| Geographic Analysis | Geographic or cultural indicators | NER, SIM + Reasoning | 6,164 | F1 |
| Victimology Profiling | Targeted victims or attacker motives | NER, REX + Reasoning | 18,612 | F1 |
| Infrastructure Extraction | Domains, IPs, URLs, or file hashes | NER, REX, SUM + Reasoning | 24,129 | F1 |
| Actor Identification | The threat group or actor (e.g., APT28) | NER, RAG, MAP + Reasoning | 17,823 | F1 |
| Campaign Correlation | Threat campaigns or incidents | NER, MAP + Reasoning | 27,762 | F1 |
| *Behavior Analysis* | | | | |
| File System Activity Detection | Suspicious file creation, deletion, or access | SPA, NER, SUM + Reasoning | 4,653 | Sim |
| Network Behavior Profiling | Patterns of external communication (e.g., C2) | SPA, NER, SUM + Reasoning | 2,617 | Sim |
| Credential Access Detection | Theft or misuse of credentials | SPA, NER, SUM + Reasoning | 2,492 | Sim |
| Execution Context Analysis | Execution behaviors by user or process | SPA, NER, SUM + Reasoning | 23,888 | Sim |
| Command & Script Analysis | Suspicious commands or scripts | SPA, NER, SUM + Reasoning | 20,232 | F1 |
| Privilege Escalation Inference | Privilege escalation attempts | SPA, NER, SUM + Reasoning | 15,953 | Sim |
| Evasion Behavior Detection | Evasion or obfuscation techniques | SPA, NER, SUM + Reasoning | 8,973 | Sim |
| Event Sequence Reconstruction | Timeline of attack-related events | SUM + Reasoning | 23,265 | Sim |
| TTP Extraction | Tactics, techniques, and procedures | RAG, MAP + Reasoning | 28,292 | F1 |
| *Prioritization* | | | | |
| Attack Vector Classification | Exploitation vectors (e.g., network, local, physical) | SUM, CLS + Reasoning | 17,448 | Acc |
| Attack Complexity Classification | Level of hurdles required to carry out the attack | SUM, CLS + Reasoning | 17,116 | Acc |
| Privileges Requirement Detection | Level of access privileges an attacker needs | SUM, CLS + Reasoning | 18,030 | Acc |
| User Interaction Categorization | If exploitation requires user participation | SUM, CLS + Reasoning | 17,075 | Acc |
| Attack Scope Detection | If the vulnerability affects one/multiple components | SUM, CLS + Reasoning | 18,570 | Acc |
| Impact Level Classification | Consequences on confidentiality, integrity, and availability | SUM, CLS + Reasoning | 18,736 | Acc |
| Severity Scoring | A numerical score indicating the overall attack severity | SUM, MATH + Reasoning | 11,507 | Dist |
| *Response & Mitigation* | | | | |
| Playbook Recommendation | Relevant response actions based on threat type | RAG, SUM + Reasoning | 10,718 | Hit |
| Security Control Adjustment | Firewall rules, EDR settings, or group policies | RAG, SUM + Reasoning | 9,929 | Sim |
| Patch Code Generation | Code snippets to patch the vulnerability | RAG, SUM + Reasoning | 11,341 | Pass |
| Patch Tool Suggestion | Security tools or utilities | RAG, SUM + Reasoning | 9,763 | Hit |
| Advisory Correlation | Security advisories or best practices | RAG, SUM + Reasoning | 24,511 | Hit |

textual metadata. We aggregate threat entries from established sources such as NVD (National Institute of Standards and Technology (NIST), 2024), MITRE CVE (The MITRE Corporation, n.d.), ATT&CK (MITRE Corporation, 2024), CWE (MITRE Corporation, b), CAPEC (MITRE Corporation, a), D3FEND (MITRE Corporation, c), Exploit-DB (Offensive Security, 2024), and VulDB (VulDB Team, 2024). These sources include detailed insights such as exploitability scores (EPSS (FIRST, b)), severity metrics (CVSS (FIRST, a)), and remediation guidance. Additionally, we incorporate data from vendor-maintained repositories (e.g., the Microsoft Security Update Guide (Microsoft Corporation), IBM X-Force (IBM Corporation, 2024)) to capture fine-grained details on affected systems, attack vectors, and patch methods.

**Threat intelligence platforms** complement these databases by providing behavioral and contextual signals linked to adversary activity. Platforms such as VirusTotal (VirusTotal, 2024), AlienVault OTX (CISA, 2024), and MISP (MISP Project, 2024) contribute indicators of compromise (IOCs), behavioral patterns, and telemetry that enable tasks

like campaign correlation, infrastructure extraction, and actor attribution. Furthermore, industry threat reports—from sources, such as Mandiant (Mandiant (Google Cloud), 2024), Recorded Future (Recorded Future, 2024), Palo Alto Unit 42 (Palo Alto Networks, 2024), and Apache (The Apache Software Foundation, 2024), offer semi-structured intelligence, including incident timelines, IOC lists, and narrative analyses, which are essential for modeling multi-stage attack sequences and evaluating blue team responses.

Additional details on how these databases and platforms are used are provided in Appendix A.

### 3.3. Standardized Threat Hunting with Operational Modules

**Task Dependency.** Threat hunting is inherently a multi-stage analytical process (Caltagirone et al., 2013), where downstream actions, such as incident response and mitigation, rely on outcomes derived from upstream analytical steps. For example, recommending an effective response playbook requires accurate attribution of the threat actor

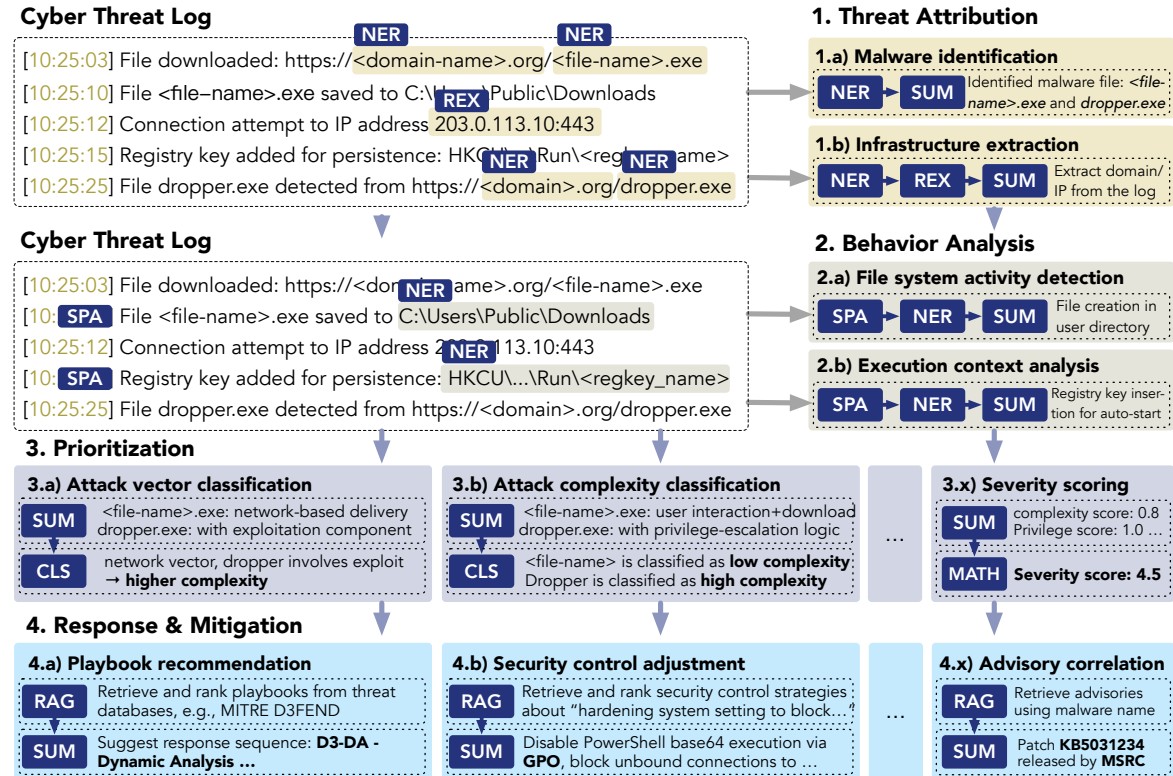

*Figure 2.* A threat-hunting example showing a dependency chain of analytical tasks, where each task is autonomously completed through reasoning (omitted for simplicity) and operational modules executed by LLMs.

and thorough behavioral analysis of the compromise. To explicitly model this structured workflow, CYBERTEAM formulates threat hunting as a *Dependency Chain*. As illustrated in Figure 2, all analytical tasks (e.g., 1.a: Malware Identification or 2.a: File System Activity Detection) are organized into a pipelined workflow that reflects their inherent dependencies. For example, attack complexity classification relies on prior analyses of file system activity and execution context. Meanwhile, tasks within the same category (e.g., malware identification and infrastructure extraction under threat attribution) can often be performed in parallel, as they address distinct dimensions of the threat and do not exhibit direct interdependencies.

> **Highlight 💡.** In CYBERTEAM, **LLMs are asked to determine which tasks to perform** at each stage, which retains reasoning flexibility in threat hunting.

**Operational Modules.** Within each threat hunting task, CYBERTEAM invokes a set of operations (functional modules) designed to produce actionable threat analyses and progressively address the current threat hunting target (e.g., incident response). Specifically, each threat hunting task $t_i$ is associated with a corresponding set of operational modules $\mathcal{F}_i = \{f_i^1, f_i^2, \dots\}$. Each task $t_i$ involves executing a sequence $f_i^* \in \mathcal{F}_i$, as detailed in the third column of

Table 2. The resulting output $y_i = f_i^*(x)$ is subsequently passed to dependent downstream tasks. For instance, the task of *TTP Extraction* involves invoking both Retrieval-Augmented Generation (RAG) and Mapping (MAP) functions to identify relevant tactics and techniques from unstructured logs. Subsequently, a downstream task such as *Tool Suggestion* utilizes RAG and summarization (SUM) functions to map these identified TTPs to suitable defensive tools.

> **Highlight 💡.** These modules provide **broad coverage of threat hunting practices** (as shown in Table 2), while retaining flexibility (e.g., in SUM, RAG) for LLM reasoning to adapt across diverse scenarios, thereby **balancing flexibility with standardization** in threat hunting.

Due to space constraints, we defer implementation details and design rationales to Appendix B.

## 4. Experiment

CYBERTEAM aims to empirically address the following research questions: **RQ₁:** How effective is standardization compared to open-ended reasoning for threat-hunting tasks? **RQ₂:** Can LLMs accurately solve individual threat-hunting tasks? **RQ₃:** How robust are LLMs, under the guidance of

CYBERTEAM, when analyzing noisy inputs?

**LLMs.** We evaluate a range of industry-leading large language models, including Grok-4.1 (GK), GPT-5.1 (G5), Qwen3-32B (QW), Gemini-3 (GM), Claude-Sonnet-4 (CD), Llama-3.1-405B (L3.1), Llama-4-Scout-17B (L4), and Gemma-3-27b (GA). In addition, we assess state-of-the-art cybersecurity-focused LLM agents, including Lily-Cybersecurity-7B (LY) (Segolily Labs, 2025), DeepHat-7B (DH) (DeepHat, 2025), and SevenLLM-7B (SL) (Ji et al., 2024).

**Open-ended Reasoning.** In open-ended reasoning, we consider three widely used prompting structures: (1) In-Context Learning (ICL) (Dong et al., 2024) – including basic task instructions along with five (or ten) illustrative examples to demonstrate the desired solution format. (2) Chain-of-Thought (CoT) (Wei et al., 2022) – encouraging the model to generate "step-by-step" reasoning results before producing the final answer. (3) Tree-of-Thought (ToT) (Yao et al., 2023) – guiding LLMs to explore multiple reasoning paths and select the most plausible one.

**Metrics.** Table 2 lists evaluation metrics tailored to each task. For information extraction tasks (e.g., malware identification), we use the **F1 score** to balance precision and recall. For classification tasks (e.g., privilege escalation inference), we adopt **accuracy** among well-defined categories. Generation or summarization tasks (e.g., behavioral profiling) are evaluated using **BERTScore** (Zhang et al., 2020) to measure semantic similarity. Tasks involving ranking (e.g., security playbook recommendation) utilize **Hit@k** (default $k = 10$), measuring whether correct choices appear in the top-k outputs. For programmatic outputs (e.g., patch code generation), we apply **Pass** rate using UNITEST in Python to assess functional correctness. Numeric estimation tasks (e.g., severity scoring) are evaluated using **Normalized Distance** to quantify similarity to ground truth values. All metrics are scaled to the range [0, 1]. We explain the rationale for those metrics in Appendix C.

### 4.1. Standardized Threat Hunting vs. Open-Ended Reasoning (RQ₁)

Ultimately, CYBERTEAM is designed to generate actionable responses and mitigation strategies against cyber threats. We begin by evaluating the overall quality of LLM-generated responses and mitigation outputs on CYBERTEAM. Table 3 presents the results, using task-specific metrics detailed in Table 2.

**Effectiveness of Standardization.** From Table 3, we observe that using operational modules (**Ours**) outperforms typical open-ended reasoning methods. For instance, modular operations enable Grok to achieve over 90% Hit@10 in playbook recommendation and over 92% in advisory correlation. In contrast, open-ended reasoning achieves only secondary effectiveness, with a significant performance gap observed (e.g., in the security control adjustment task of SevenLLM). This demonstrates the effectiveness of combining standardized guidance with the inherent flexibility of LLMs.

**Gains from Standard Operating Procedures.** Notably, while ICL, CoT, and ToT have been shown to improve generation quality for general-purpose tasks (Dong et al., 2024; Wang et al., 2023), they provide limited guidance for domain-specific problems that require precise procedural knowledge and structured analytical workflows. By contrast, standardized threat hunting workflows help LLMs follow standard operating procedures by decomposing complex tasks into modular steps. This reduces hallucination and enforces structure. In tasks requiring strict sequencing (e.g., threat actor identification followed by response planning), workflow-based methods ensure the correct order and information flow, outperforming ICL, CoT, and ToT, which often lack such control.

> **Case Study I** (Failure Case). When using CoT to generate a response plan for LockBit (a ransomware), GPT-5.1 offers generic recommendations *"... the first step is to isolate affected machines. Next, the system should assess backup availability and notify stakeholders ..."* without tailoring to LockBit and ignoring unique traits like double extortion tactics or known exploits.

By contrast, operations in CYBERTEAM constrain LLM reasoning to resolve correct analytical sequences, ensuring outputs remain aligned with operational goals:

> **Case Study II** (Successful Case). The modular operation framework guides GPT-5.1 to explicitly invoke **RAG** and **SUM** modules. Specifically, RAG retrieves up-to-date security advisories (e.g., *CISA Alert AA23-325A*) specific to LockBit, while SUM outlines mitigation strategies with *double extortion prevention* and *air-gapped offline backups*.

These results suggest that in cybersecurity, particularly in threat-hunting scenarios, structured reasoning methods are necessary for reliably leveraging LLM capabilities.

**Interpretability in Operations.** Notably, the modular approach enhances interpretability for analysts, as outputs can be traced back to specific operations (e.g., RAG for evidence retrieval, SUM for summarization). In contrast, open-ended prompts produce opaque reasoning chains that are harder to audit what real-world evidence is integrated.

*Table 3.* Results of LLMs threat-hunting performance (scaled to 100%) on CYBERTEAM, using corresponding metrics tailored to each analytical target as detailed in Table 2. We use **boldface** to indicate the best results and underline to denote the second-best results.

| Method | | Cybersecurity Agent | | | Industry-Leading LLM | | | | | | | |
|---|---|---|---|---|---|---|---|---|---|---|---|---|
| | | LY | DH | SL | GK | G5 | QW | GM | CD | L3.1 | L4 | GA |
| **Playbook Recommend** | | | | | | | | | | | | |
| Open-ended | ICL5 | $42.3_{\pm1.9}$ | $54.2_{\pm2.4}$ | $54.7_{\pm2.3}$ | $65.8_{\pm2.6}$ | $74.4_{\pm2.8}$ | $52.8_{\pm2.0}$ | $79.4_{\pm1.8}$ | $63.7_{\pm2.1}$ | $65.8_{\pm2.6}$ | $55.8_{\pm2.0}$ | $54.9_{\pm2.1}$ |
| | ICL10 | $44.1_{\pm1.8}$ | $52.5_{\pm2.6}$ | $55.3_{\pm2.7}$ | $66.5_{\pm2.7}$ | $75.8_{\pm2.7}$ | $53.6_{\pm2.1}$ | $80.2_{\pm1.9}$ | $64.9_{\pm2.2}$ | $66.4_{\pm2.7}$ | $56.4_{\pm2.1}$ | $55.5_{\pm2.2}$ |
| | CoT | $51.6_{\pm2.2}$ | $50.6_{\pm2.4}$ | $50.5_{\pm2.2}$ | $79.6_{\pm1.8}$ | $90.5_{\pm1.7}$ | $67.5_{\pm2.5}$ | $80.1_{\pm1.6}$ | $81.4_{\pm1.7}$ | $77.3_{\pm2.0}$ | $67.3_{\pm2.3}$ | $66.4_{\pm2.2}$ |
| | ToT | $48.1_{\pm2.4}$ | $53.3_{\pm2.6}$ | $54.3_{\pm2.5}$ | $76.5_{\pm2.1}$ | $86.4_{\pm1.8}$ | $71.4_{\pm2.4}$ | $83.5_{\pm1.7}$ | $77.2_{\pm2.0}$ | $82.1_{\pm1.8}$ | $72.1_{\pm2.1}$ | $71.2_{\pm2.2}$ |
| **Standardized (Ours)** | | $67.2_{\pm1.7}$ | $58.4_{\pm2.1}$ | $66.8_{\pm1.9}$ | $85.9_{\pm1.4}$ | $92.7_{\pm1.3}$ | $79.3_{\pm2.0}$ | $91.8_{\pm1.3}$ | $89.3_{\pm1.6}$ | $89.7_{\pm1.5}$ | $79.7_{\pm1.9}$ | $78.8_{\pm2.0}$ |
| **Security Control Adjust** | | | | | | | | | | | | |
| Open-ended | ICL5 | $51.5_{\pm2.4}$ | $66.3_{\pm2.3}$ | $43.9_{\pm2.7}$ | $63.1_{\pm2.5}$ | $71.6_{\pm0.8}$ | $50.6_{\pm2.2}$ | $65.8_{\pm2.1}$ | $79.2_{\pm1.9}$ | $61.5_{\pm2.4}$ | $51.5_{\pm2.0}$ | $50.6_{\pm1.1}$ |
| | ICL10 | $53.2_{\pm2.5}$ | $68.4_{\pm2.4}$ | $45.6_{\pm2.8}$ | $64.0_{\pm2.1}$ | $73.1_{\pm2.7}$ | $51.2_{\pm0.1}$ | $66.4_{\pm2.2}$ | $80.1_{\pm1.8}$ | $62.3_{\pm2.4}$ | $52.3_{\pm2.1}$ | $51.4_{\pm2.1}$ |
| | CoT | $60.3_{\pm2.0}$ | $70.5_{\pm2.1}$ | $68.4_{\pm1.9}$ | $71.6_{\pm1.9}$ | $81.5_{\pm1.7}$ | $59.8_{\pm2.4}$ | $79.2_{\pm1.7}$ | $77.2_{\pm1.9}$ | $77.9_{\pm1.8}$ | $67.9_{\pm1.4}$ | $63.0_{\pm2.1}$ |
| | ToT | $66.7_{\pm1.9}$ | $72.1_{\pm2.0}$ | $61.6_{\pm2.2}$ | $77.2_{\pm1.7}$ | $86.9_{\pm0.9}$ | $66.3_{\pm2.3}$ | $73.6_{\pm2.1}$ | $73.1_{\pm2.0}$ | $72.8_{\pm0.3}$ | $62.8_{\pm2.3}$ | $61.9_{\pm2.2}$ |
| **Standardized (Ours)** | | $74.2_{\pm1.6}$ | $77.6_{\pm1.5}$ | $80.1_{\pm1.7}$ | $83.4_{\pm1.5}$ | $91.0_{\pm1.1}$ | $74.7_{\pm1.8}$ | $88.5_{\pm1.5}$ | $86.5_{\pm0.7}$ | $86.4_{\pm1.7}$ | $76.4_{\pm1.8}$ | $75.5_{\pm0.9}$ |
| **Patch Code Generation** | | | | | | | | | | | | |
| Open-ended | ICL5 | $10.8_{\pm1.0}$ | $49.8_{\pm3.1}$ | $29.2_{\pm2.7}$ | $57.5_{\pm2.7}$ | $59.7_{\pm1.5}$ | $39.3_{\pm2.9}$ | $63.7_{\pm2.2}$ | $47.5_{\pm2.5}$ | $49.2_{\pm0.6}$ | $39.2_{\pm2.9}$ | $38.3_{\pm3.8}$ |
| | ICL10 | $12.6_{\pm1.6}$ | $51.2_{\pm3.0}$ | $31.5_{\pm2.8}$ | $59.1_{\pm2.6}$ | $60.4_{\pm1.4}$ | $40.1_{\pm1.8}$ | $64.9_{\pm0.3}$ | $48.6_{\pm0.4}$ | $50.1_{\pm1.1}$ | $40.1_{\pm1.7}$ | $39.2_{\pm2.7}$ |
| | CoT | $24.5_{\pm2.2}$ | $54.7_{\pm2.4}$ | $55.1_{\pm2.1}$ | $59.7_{\pm2.3}$ | $77.6_{\pm1.8}$ | $54.7_{\pm2.3}$ | $65.3_{\pm2.0}$ | $66.3_{\pm2.0}$ | $67.4_{\pm1.8}$ | $57.4_{\pm2.1}$ | $51.5_{\pm2.2}$ |
| | ToT | $25.3_{\pm2.1}$ | $50.9_{\pm2.5}$ | $58.3_{\pm2.0}$ | $63.1_{\pm2.2}$ | $73.8_{\pm1.9}$ | $50.2_{\pm2.5}$ | $69.8_{\pm1.9}$ | $61.4_{\pm2.1}$ | $62.9_{\pm2.0}$ | $52.9_{\pm2.3}$ | $52.2_{\pm2.3}$ |
| **Standardized (Ours)** | | $29.7_{\pm1.9}$ | $63.4_{\pm1.8}$ | $60.2_{\pm2.0}$ | $73.8_{\pm1.6}$ | $88.7_{\pm1.2}$ | $65.4_{\pm2.0}$ | $82.6_{\pm1.4}$ | $79.2_{\pm1.6}$ | $80.6_{\pm1.5}$ | $70.6_{\pm1.8}$ | $69.7_{\pm1.9}$ |
| **Patch Tool Suggestion** | | | | | | | | | | | | |
| Open-ended | ICL5 | $48.2_{\pm2.6}$ | $65.2_{\pm2.3}$ | $61.5_{\pm2.5}$ | $70.2_{\pm2.1}$ | $80.7_{\pm1.8}$ | $59.2_{\pm2.4}$ | $74.1_{\pm2.1}$ | $68.5_{\pm2.3}$ | $70.3_{\pm2.3}$ | $60.3_{\pm2.4}$ | $59.4_{\pm2.6}$ |
| | ICL10 | $49.1_{\pm2.5}$ | $64.7_{\pm2.4}$ | $63.1_{\pm2.6}$ | $71.0_{\pm2.1}$ | $81.9_{\pm1.8}$ | $60.3_{\pm2.4}$ | $74.9_{\pm2.0}$ | $69.8_{\pm2.2}$ | $71.4_{\pm2.2}$ | $61.4_{\pm2.4}$ | $60.5_{\pm2.5}$ |
| | CoT | $53.6_{\pm2.2}$ | $70.1_{\pm2.1}$ | $77.2_{\pm1.8}$ | $80.5_{\pm1.6}$ | $91.4_{\pm1.4}$ | $70.3_{\pm2.0}$ | $81.7_{\pm1.6}$ | $79.1_{\pm1.8}$ | $79.6_{\pm1.9}$ | $69.6_{\pm2.0}$ | $68.7_{\pm2.1}$ |
| | ToT | $56.5_{\pm2.0}$ | $71.8_{\pm2.0}$ | $68.1_{\pm2.2}$ | $77.1_{\pm1.8}$ | $87.6_{\pm1.6}$ | $74.5_{\pm1.9}$ | $86.3_{\pm1.5}$ | $83.7_{\pm1.2}$ | $84.2_{\pm1.6}$ | $74.2_{\pm1.8}$ | $67.3_{\pm2.2}$ |
| **Standardized (Ours)** | | $69.1_{\pm1.8}$ | $76.5_{\pm1.7}$ | $77.7_{\pm1.7}$ | $88.7_{\pm1.3}$ | $98.2_{\pm1.1}$ | $83.6_{\pm1.6}$ | $93.2_{\pm1.3}$ | $91.2_{\pm1.5}$ | $92.1_{\pm1.4}$ | $82.1_{\pm1.6}$ | $81.2_{\pm1.7}$ |
| **Advisory Correlation** | | | | | | | | | | | | |
| Open-ended | ICL5 | $21.7_{\pm3.8}$ | $57.5_{\pm2.6}$ | $63.8_{\pm2.5}$ | $66.0_{\pm2.3}$ | $68.5_{\pm2.3}$ | $48.5_{\pm3.3}$ | $62.4_{\pm2.6}$ | $56.8_{\pm2.7}$ | $58.7_{\pm2.6}$ | $48.7_{\pm3.1}$ | $47.8_{\pm3.2}$ |
| | ICL10 | $22.9_{\pm3.7}$ | $59.1_{\pm2.6}$ | $64.7_{\pm2.5}$ | $67.2_{\pm2.3}$ | $69.4_{\pm2.2}$ | $49.2_{\pm3.1}$ | $63.2_{\pm2.5}$ | $58.1_{\pm2.6}$ | $59.5_{\pm1.2}$ | $49.5_{\pm3.1}$ | $48.6_{\pm3.2}$ |
| | CoT | $49.5_{\pm2.3}$ | $71.4_{\pm2.0}$ | $69.5_{\pm2.1}$ | $68.5_{\pm2.1}$ | $81.8_{\pm1.7}$ | $61.7_{\pm2.4}$ | $77.5_{\pm1.8}$ | $76.2_{\pm1.9}$ | $76.3_{\pm0.9}$ | $66.3_{\pm2.1}$ | $65.4_{\pm1.1}$ |
| | ToT | $46.8_{\pm2.3}$ | $73.2_{\pm1.9}$ | $67.2_{\pm2.2}$ | $72.1_{\pm1.9}$ | $85.5_{\pm1.6}$ | $64.8_{\pm2.2}$ | $73.1_{\pm1.9}$ | $72.5_{\pm2.0}$ | $71.8_{\pm2.1}$ | $61.8_{\pm2.3}$ | $60.9_{\pm2.3}$ |
| **Standardized (Ours)** | | $73.4_{\pm1.7}$ | $78.8_{\pm1.5}$ | $77.1_{\pm1.6}$ | $81.6_{\pm1.4}$ | $93.6_{\pm1.2}$ | $76.5_{\pm1.8}$ | $86.9_{\pm1.4}$ | $84.5_{\pm1.6}$ | $84.9_{\pm1.5}$ | $74.9_{\pm1.7}$ | $74.0_{\pm1.7}$ |

**Case Study III (Interpretability).** For the MOVEit vulnerability (CVE-2023-34362), an open-ended Qwen prompt returned only a vague recommendation ("apply vendor patches and monitor suspicious traffic"). In contrast, our pipeline invoked the **RAG** module to retrieve Progress Software's advisory and the **NER** module to extract SQL injection IOCs. This modular trace improved accuracy and enabled analysts to audit advisory steps.

Due to space constraints, we provide additional evaluation of the trade-off between latency and reliability in Appendix E.1. Our results show that the standardized threat hunting method achieves a more favorable balance compared with open-ended reasoning.

### 4.2. Individual Threat-Hunting Tasks (RQ$_2$)

Complementing Section 4.1, we also evaluate individual threat-hunting tasks prior to the response & mitigation stage, as outlined in Table 2. Figures 3 and Appendix E present the experimental results.

Observe that using standardized threat hunting consistently achieves the highest performance across all intermediate tasks. However, **the magnitude of performance gains varies across task types**. For instance, in complex reasoning tasks (e.g., Event Sequence Construction), the standardized method yields substantial improvements over open-ended reasoning strategies like CoT and ToT, boosting accuracy by over 20% using GPT-5.1. These gains are most notable when task dependencies are strong. For example, generating effective responses depends on accurate upstream analysis. Meanwhile, ICL/CoT/ToT often fail to reliably solve such multi-stage tasks. In contrast, for identification-focused tasks (e.g., privilege escalation inference), the performance gap between operational modules and baseline prompting is smaller since tasks are more self-contained, and models can often arrive at correct predictions even without intensive task decomposition and reasoning.

Due to space constraints, we provide complementary results and analyses in Appendix E.2.

### 4.3. LLM Robustness against Noisy Inputs (RQ$_3$)

**Experimental Setting.** We also investigate LLM robustness when input threat logs contain noisy text. We introduce (i) token-level noise using TextAttack (Morris et al.,

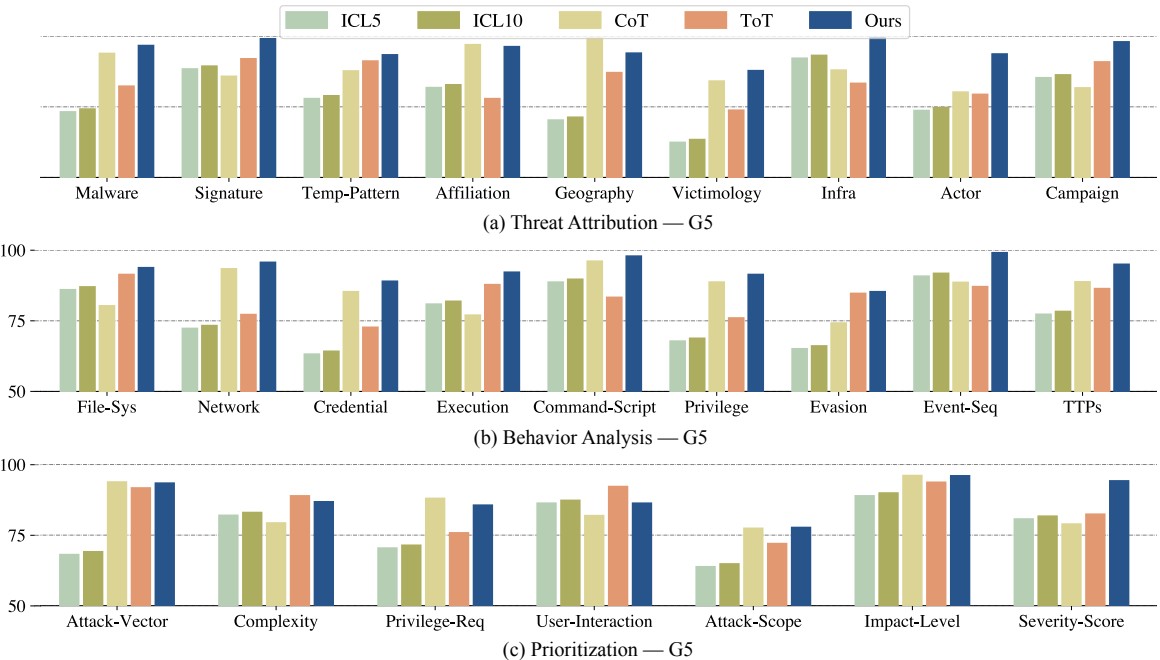

*Figure 3.* Threat-hunting performance (scaled to 100%) on individual tasks. Results for additional LLMs are provided in Appendix E.

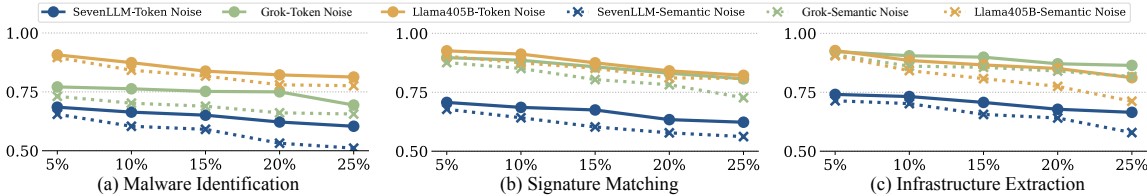

*Figure 4.* LLM performance (metrics corresponding to Table 2) when input threat logs are perturbed with token-level noise (solid line) or semantic-level noise (dashed line). X-axis shows the noise ratios.

2020), which randomly injects or substitutes tokens, and (ii) semantic-level noise using BART-paraphraser (Lewis et al., 2020), which subtly introduces misleading or shifted context. Both noise types are applied at controlled levels (e.g., perturbing 10% of the input).

**Results and Observations.** From Figure 4, we observe that token-level noise has a smaller impact on LLM performance compared to semantic-level noise. For example, under 10% perturbation, random character insertions or deletions lead to less than 5% performance drop across tasks. In contrast, semantic-level noise (e.g., paraphrased or subtly altered context) causes a much larger decline. These findings suggest that while LLMs handle surface-level errors relatively well, they struggle with the semantic shifting, even when guided by CYBERTEAM. This highlights the importance of curating expert-level threat reports in threat hunting, as imprecise statements can unintentionally mislead blue team efforts and degrade overall analysis.

### 4.4. Additional Experiments

Due to space limitation, we defer experiments on (i) running time to §E.1, (ii) human evaluation to §E.3, (iii) ablation study to §E.4, (iv) token consumption to §E.5, and (v) comparison to non-LLM baselines to §E.6.

## 5. Conclusion

We present CYBERTEAM, a benchmark designed to evaluate the capabilities of LLMs in blue-team threat hunting. CYBERTEAM integrates broad real-world datasets, standardized workflows, and detailed evaluation strategies, thus providing a comprehensive investigation for LLMs' capabilities in realistic blue-team scenarios. Our empirical results offer actionable insights for integrating standardized operations into security workflows. We hope CYBERTEAM will serve as a valuable resource for the research community for future innovations in AI-assisted cybersecurity.

## Impact Statement

This paper introduces a benchmark designed to improve how AI models help defend against cyberattacks. Our work creates a standardized way for AI models to analyze threats and aims to make digital environments safer for organizations and individuals. We recognize that while these tools are built for defense, there is a risk that the methods used to improve AI reasoning could be studied by others to find new ways to bypass security. However, we believe the benefit of providing defenders with more reliable, automated tools to stop complex attacks far outweighs these risks. Our research encourages the development of more transparent and helpful AI for cybersecurity, ultimately helping society keep pace with the growing number of online threats.

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

# Appendix

**Table of Contents**

# A. Data Source and Metadata Collection

## A.1. Data Source

**The MITRE CVE (Common Vulnerabilities and Exposures)** system (The MITRE Corporation, n.d.) is a foundational database that provides unique identifiers for publicly disclosed cybersecurity vulnerabilities. Each CVE record includes an ID, a brief description, references to external resources, and associated vendors or platforms. This source allows for consistent naming and indexing of vulnerabilities across tools and reports. We collect structured metadata such as CVE IDs, descriptions, reference links, and related CWE classifications. CVE feeds (XML/JSON) are used for automated ingestion and linkage to other threat intelligence frameworks like CAPEC and ATT&CK.

Maintained by NIST, the **NVD (National Vulnerability Database)** (National Institute of Standards and Technology (NIST), 2024) builds on MITRE CVE data by adding rich metadata, including CVSS scores (base, temporal, environmental), CWE mappings, configuration impacts, patch availability, and severity vectors. We extract metadata through the official JSON data feeds, parsing CVE-level risk metrics, impact sub-scores, and associated product configurations. This information is critical for prioritizing remediation and understanding the real-world impact of vulnerabilities.

**Exploit-DB** (Offensive Security, 2024) is a curated collection of publicly available exploits and proof-of-concept code. Each entry includes exploit titles, CVE references, author information, platform tags, and the actual code used in attacks. Unlike CVE/NVD, Exploit-DB provides practical insights into how vulnerabilities are weaponized in real environments. We extract titles, descriptions, exploit types (e.g., Local, Remote), and related CVEs using web scraping and NLP-based text classification.

**CWE(Common Weakness Enumeration)** (MITRE Corporation, b) is a taxonomy developed by MITRE to classify software and hardware weaknesses. Each CWE includes a unique ID, a detailed explanation, potential consequences, examples, and related patterns (e.g., CAPEC). We use CWE to enrich CVE data with root cause information, enabling fine-grained vulnerability clustering and defensive prioritization. The metadata includes weakness category, severity, and relationships with CAPEC and CVE entries.

**CAPEC (Common Attack Pattern Enumeration and Classification)** (MITRE Corporation, a) provides a standardized catalog of common attack strategies. Each pattern includes the attacker's objectives, prerequisites, execution flow, related weaknesses (CWE), and example scenarios. We extract attack pattern IDs, descriptions, related CWEs, and suggested mitigations. These data points enable us to map vulnerabilities to adversarial behaviors, enhancing our CTI behavioral modeling capabilities.

**The MITRE ATT&CK** (MITRE Corporation, 2024) framework systematically catalogs adversary tactics, techniques, and procedures (TTPs) observed in real-world incidents. Each entry includes tactic categories (e.g., Privilege Escalation), techniques, mitigations, detection suggestions, and threat actor mappings. We extract technique IDs, corresponding software, mitigation strategies, and detection methods. These are used to link CVEs and exploits to higher-level attacker behaviors, supporting advanced threat modeling.

**D3FEND** (MITRE Corporation, c) is a curated knowledge graph that maps defensive techniques to specific threat behaviors and artifacts. D3FEND complements the well-known ATT&CK framework by focusing on how defenders can detect, disrupt, and respond to adversarial actions. To integrate this resource into CYBERTEAM, we crawl D3FEND's publicly available ontology and extract metadata on detection, deception, and mitigation techniques, along with their associated digital artifacts (e.g., file paths, registry keys, network signatures). This metadata is then linked to relevant analytical tasks, such as behavioral profiling and response planning, providing a rich, standardized reference for grounding LLM outputs in practical defensive actions.

**Oracle Security Alerts** (Oracle Corporation, 2024) provides detailed security patch advisories for its product suite. Each alert includes the CVEs addressed, severity scores, and remediation timelines. We parse the advisories to gather product-specific vulnerability timelines, vendor patch statuses, and mitigation instructions, which complement the NVD and MITRE CVE datasets.

**Red Hat Bugzilla** (Red Hat, Inc., 2024) is a bug tracking system that includes detailed discussions and technical logs about software bugs, many of which are security-related. Entries often include CVE links, fix status, patch availability, and affected components. We scrape metadata such as Bug IDs, CVE references, affected packages, and resolution details to supplement our understanding of vulnerability lifecycle management.

**The RHSA(Red Hat Security Advisories)** (Red Hat, Inc.) portal lists all critical, important, and moderate security advisories affecting Red Hat products. Each advisory provides CVE mappings, severity scores, fixed packages, and risk summaries. Metadata extraction includes advisory IDs, publication dates, CVE linkages, and suggested upgrades or patches, enabling alignment with real-world remediation practices.

**IBM X-Force Exchange** (IBM Corporation, 2024) is a commercial threat intelligence sharing platform that provides in-depth reports on vulnerabilities, exploits, malware, and threat actors. Each CVE entry is enriched with exploitability status, malware connections, and actor attribution. We extract structured threat metadata such as exploit availability, indicators of compromise (IOCs), campaign tags, and actor profiling to complement CVE risk modeling.

**CISE (Cybersecurity Information Sharing Environment)** (CISE Program, 2024), maintained by CISA, promotes cybersecurity information exchange across government and private sector entities. The platform facilitates sharing of indicators of compromise (IOCs), analysis reports, and threat mitigation strategies through structured partnerships. We extract strategic-level threat metadata, including threat vectors, vulnerability trends, and response best practices from shared reports and alerts. This supports broader CTI tasks like attribution and risk contextualization.

**VulDB (Vulnerability Database)** (VulDB Team, 2024) is a commercial vulnerability intelligence service that provides insights into current exploits, threat actor behavior, and exploit trends. Entries often include exploitability scores, attack vectors, exploitation status, and tags related to malware or campaigns. We collect CVE mappings, vulnerability titles, exploitation timelines, and associated actors, enabling temporal and behavioral correlation with other sources like Exploit-DB and MITRE ATT&CK.

**Apache**'s official security advisory page lists all disclosed vulnerabilities affecting Apache projects (e.g., HTTP Server, Tomcat, Struts) (The Apache Software Foundation, 2024). Each advisory includes CVE references, affected versions, and patch instructions. We extract CVE mappings, patch details, vulnerability types, and affected modules. These insights are cross-referenced with MITRE CVE and NVD entries to improve accuracy in software-specific threat tracking.

**Mandiant Threat Intelligence Reports** (Mandiant (Google Cloud), 2024), now part of Google Cloud, publishes in-depth research on nation-state APTs, malware campaigns, and threat actor tactics. Their reports include IOC lists, ATT&CK mappings, and campaign chronologies. We extract metadata on APT groups, attack stages, observed TTPs, and malware toolkits. These data points support the attribution and behavioral modeling dimensions of our threat intelligence corpus.

**Recorded Future Threat Intelligence Reports** (Recorded Future, 2024) publishes real-time, machine-readable threat intelligence covering threat actors, vulnerabilities, dark web chatter, and geopolitical cyber campaigns. Reports often include structured indicators, predictive analytics, and CVE exploitability assessments. We leverage this source to collect threat context, emerging trends, and exploit discussion patterns—enabling our system to associate vulnerabilities with evolving threat actor intent and capability.

**Unit 42 Threat Research (Palo Alto Networks)** (Palo Alto Networks, 2024) provides malware analysis, campaign forensics, and actor behavior insights from Palo Alto Networks' global threat intelligence platform. Their publications include links to malicious infrastructure, malware families, and ATT&CK references. We extract TTPs, CVE-to-malware correlations, and campaign data. This enhances our contextual metadata for linking specific vulnerabilities to real-world exploitation scenarios.

**Microsoft's Security Update Guide** (Microsoft Corporation) lists monthly updates across its software stack. Entries contain CVEs, severity ratings, exploitability assessments, patch availability, and affected platforms. Metadata extraction includes CVE linkage, threat vectors (e.g., local, remote), exploitation likelihood, and patch rollout status—enriching vendor-specific vulnerability intelligence.

**CVSS (Common Vulnerability Scoring System)** (FIRST, a) is a widely adopted scoring system developed by FIRST to assess the severity of software vulnerabilities. It breaks down risk into Base, Temporal, and Environmental components. We use this framework to interpret NVD scores, compare severity across platforms, and calibrate exploitability in relation to business-critical systems.

**EPSS (Exploit Prediction Scoring System)** (FIRST, b), also developed by FIRST, provides probabilistic predictions of whether a vulnerability is likely to be exploited in the wild. It integrates data from CVSS, Exploit-DB, and historical attack patterns. We ingest EPSS scores via API to prioritize vulnerabilities not just by severity, but by real-world exploitation likelihood—enabling dynamic risk-based vulnerability management.

**MISP (Malware Information Sharing Platform)** (MISP Project, 2024) is an open-source platform designed for structured threat intelligence sharing using STIX/TAXII formats. It facilitates sharing of IOCs, threat event correlations, and TTP mappings. We integrate MISP data via its API to ingest indicators (e.g., hashes, domains, IPs), related threat actors, and event metadata. These enrich our knowledge graph with actionable CTI feeds.

**VirusTotal** (VirusTotal, 2024) is a widely used threat intelligence platform that aggregates malware analysis and sandbox reports from multiple antivirus engines and security vendors. To support behavior analysis and attribution tasks, CYBERTEAM collects structured threat metadata from VirusTotal's public API, including file hashes (MD5, SHA-1, SHA-256), behavioral execution traces, contacted IPs/domains, dropped files, and detection labels. This information is linked to threat artifacts such as malware families, indicators of compromise (IOCs), and known campaign signatures. The extracted metadata enables CYBERTEAM to contextualize adversarial behaviors and enrich analytical functions like malware classification, infrastructure extraction, and campaign correlation.

**AlienVault Open Threat Exchange (OTX)** (CISA, 2024) is a collaborative threat-sharing platform that provides community-contributed threat indicators and contextual threat intelligence. CYBERTEAM leverages the OTX API to collect threat pulses—curated collections of IOCs and metadata describing specific threat actors, campaigns, or vulnerabilities. These pulses include information such as associated IPs, domains, file hashes, CVEs, and targeted sectors. By integrating OTX data, CYBERTEAM enhances its ability to support tasks like actor attribution, TTP matching, and community correlation, allowing LLMs to reason over shared intelligence and align analysis with ongoing threat landscapes.

**Data Ethics.** All data used in CYBERTEAM are collected from publicly available vulnerability databases and open-source threat intelligence platforms. No sensitive personal information or proprietary organizational data are included.

### A.2. Data Preparation

**Collection & Processing Settings.** Our benchmark integrates threat intelligence from two categories of sources: (1) structured vulnerability databases and (2) semi-structured threat intelligence platforms. For the former, we collect machine-readable feeds from authoritative repositories such as NVD, MITRE CVE, CWE, CAPEC, D3FEND, Exploit-DB, VulDB, and EPSS. These sources provide canonical identifiers (e.g., CVE-ID), structured metadata (e.g., CVSS vectors, exploit maturity, CWE type), and standardized taxonomies that serve as grounding for downstream tasks. For the latter, we ingest reports, advisories, and campaign summaries from industry platforms including VirusTotal, AlienVault OTX, Mandiant, Unit 42, IBM X-Force, Cisco Talos, and Recorded Future. These sources contain higher-level context such as TTPs, indicators of compromise (IoCs), adversary attribution, and remediation actions. All raw sources undergo normalization (UTF-8 encoding, HTML stripping) before task-specific extraction, during which we decompose documents into evaluation units aligned with one of the five defined threat-hunting tasks. This pipeline ensures that the benchmark combines comprehensive structured knowledge with real-world narrative intelligence.

**Validation & Quality Control.** We apply a multi-stage validation pipeline to ensure the reliability and semantic consistency of all benchmark entries. Structured vulnerability feeds undergo schema validation to guarantee the completeness and format correctness of required fields (e.g., CVSS vectors must contain all eight base metrics; exploit maturity must match predefined categorical values). For threat reports, we apply heuristic validation steps such as cross-referencing CVE identifiers with NVD, enforcing ATT&CK technique format compliance (Txxxx), and verifying timestamp consistency. Additionally, we perform cross-source coherence checks to ensure that metadata reported by intelligence platforms matches the underlying vulnerability or campaign identifiers when available. To complement automated validation, human verification is applied to sampled entries from each source category to confirm contextual accuracy and to remove non-actionable or irrelevant content (e.g., marketing or promotional blog posts). This layered validation approach guarantees that all benchmark samples are both structurally valid and operationally meaningful.

**Deduplication.** Because multiple sources often report the same underlying security event, we implement deduplication at three levels. First, structural deduplication collapses entries that reference the same canonical identifier (e.g., CVE-ID, Exploit-DB ID, Malware hash) into a single base record that consolidates metadata across sources. Second, behavioral deduplication removes near-duplicate threat reports by computing MinHash signatures and merging samples whose Jaccard similarity exceeds 0.85, ensuring that mirrored advisories or lightly reworded vendor posts do not inflate dataset size or bias evaluations. Third, we apply task-instance deduplication after report decomposition: if a single report contributes to multiple task categories, each instance is retained exactly once per task, and the final data statistics reflect unique evaluation units rather than repeated documents. This deduplication strategy prevents redundancy while preserving the richness of multi-perspective threat intelligence.

## B. Modularized Operations: Basic Component of Standardized Threat Hunting

To support modular and extensible capabilities within our CYBERTEAM, we decompose complex NLP workflows into discrete, modularized operations. This section detail the implementation of NLP modules as described in section 3.1. Each module corresponds to a specific operation type, described as follows:

### B.1. NER (Named Entity Recognition)

To identify and classify cybersecurity-relevant entities such as threat actors, malware names, vulnerabilities, and indicators of compromise (IOCs) in unstructured textual data, NER facilitates automated extraction for threat attribution and situational awareness. We employ prompt-based techniques that enable entity recognition without retraining, thus maintaining adaptability to emerging domain vocabulary, specifically:

**Implementation.** At execution time, the agent calls NER through inputs containing the raw threat events (e.g., log snippet, CTI fragment), a threat-hunting stage identifier (e.g., attribution, behavior analysis, response planning), or a configurable entity schema. The LLM is then instructed to return a structured JSON-compatible object that adheres to this schema. To enforce reliability, the function is wrapped with automatic retry logic and schema validation; if the model returns malformed fields or missing keys, the system invokes a repair cycle using a lightweight self-correction prompt. This keeps the module robust to hallucinations while avoiding the cost of supervised fine-tuning.

---

**Prompt Template: NER for Cyber Threat Hunting**

**System Prompt:**
You are a cybersecurity threat-hunting assistant with expertise in extracting structured security intelligence from noisy or unstructured text. Your task is to identify and categorize all cybersecurity-relevant entities present in the input and return them in a machine-readable format.
**Instructions:** Given the input text, extract all relevant named entities and classify them into the following categories (include only those that appear):

- **Threat Actor:** Names or labels of attacker groups or individuals (e.g., APT28, TA505, Lazarus).

- **Malware / Tool:** Malware families, implants, exploits, payloads, living-off-the-land tools (e.g., Cobalt Strike, Mimikatz).

- **Vulnerability:** CVE identifiers, zero-days, protocol flaws.

- **Infrastructure / IOC:** IPs, domains, URLs, file hashes, registry keys, cloud resources, container images.

- **Technique / TTP:** MITRE ATT&CK IDs, exploitation patterns, privilege escalation, lateral movement.

- **Target / Asset:** Affected platforms, operating systems, cloud accounts, business units.

- ...<Exhaustive list omitted>...

**Output Format:**
Return a JSON object with the following structure:

```
{
  "threat_actor": [...],
  "malware_or_tool": [...],
  "vulnerability": [...],
  "infrastructure": [...],
  "technique_or_ttp": [...],
  "target_or_asset": [...],
  ...<Exhaustive list omitted>...
}
```

Each value must be a list. Use empty lists if no entities are found.
**Important Notes:**

---

- Do not infer or hallucinate entities not grounded in the text.

- Normalize formats (e.g., lowercase domains, uppercase CVEs).

- Remove duplicates and irrelevant entities.

- Do not include explanations or reasoning in the output.

**User Input:**
<INSERT TEXT HERE>

**Design Rationale:** In real-world threat hunting, analysts are constantly overwhelmed by unstructured reports, logs, and advisories filled with technical jargon and entity references. Automating named entity recognition helps blue teams immediately isolate critical items such as threat actors, malware strains, or CVE identifiers without combing through entire reports manually. This reduces analyst workload, accelerates attribution, and ensures no important entity slips through, particularly when adversaries recycle or slightly modify names and indicators across campaigns.

## B.2. REX (Regex Parsing)

The Regex Parsing (REX) module is responsible for extracting syntactically well-defined threat indicators from unstructured logs, reports, emails, or telemetry feeds using a library of predefined regular expressions. Unlike semantic NER, which delegates pattern understanding to a language model, REX enforces high-precision, rule-based extraction for entities that follow strict syntactic formats such as IPv4/IPv6 literals, domain names, URLs, file hashes (MD5/SHA1/SHA256), registry paths, email addresses, timestamps, MITRE ATT&CK IDs, and CVE identifiers. This makes REX particularly well suited for early-stage pipeline normalization, IOC pivoting, forensic processing, and cross-document correlation.

**Implementation**. The REX module is deployed as a lightweight function-wrapped operator inside the agent runtime. When invoked, the LLM is not responsible for pattern recognition itself; instead, the model triggers a function containing a curated regex registry indexed by indicator type. These patterns are compiled once at initialization using Python's re module or Rust-based regex engines for performance. The input text is streamed through a parsing pipeline, where each regex class is applied either sequentially or in parallel depending on configured performance constraints. Matches are normalized, deduplicated, and stored in a structured return schema that downstream modules (e.g., RAG or MAP) can consume.

**Prompt Template: Regex-Based Indicator Extraction**

**System Prompt:**
You are a cybersecurity log parsing assistant. Your task is to extract structured threat indicators using deterministic pattern matching (regex). Do not infer or guess values—only return data that matches well-formed patterns.
**Instructions:** Given the text below, extract all instances matching the following indicator classes:

- **IP Address:** IPv4 or IPv6 literals

- **Domain / URL:** Fully qualified domain names or URLs

- **File Hash:** MD5, SHA1, SHA256

- **CVE Identifier:** Format CVE-YYYY-NNNNN

- **Timestamp:** Any ISO, syslog, Windows, or common log format

- **ATT&CK Technique:** Format TXXXX[.XXX]

- ...<Exhaustive list omitted>...

**Output Format:**
Return a JSON object in the following schema:

```
{
  "ip": [...],
```

```
    "domain": [...],
    "url": [...],
    "file_hash": [...],
    "cve": [...],
    "timestamp": [...],
    "attack_technique": [...],
    ...<Exhaustive list omitted>...
}
```

**Rules:**

- Output only syntactically valid matches.

- Do not include any reasoning or natural language explanation.

- Normalize formats (e.g. lowercase domains, uppercase CVEs, uppercase hashes).

- Return empty lists for categories with no matches.

- Deduplicate all results.

**User Input:**
<INSERT TEXT HERE>

**Fault Tolerance.** The REX function supports runtime extension, enabling operators to dynamically inject new patterns or adjust matching rules without retraining any models. In cases where extraction ambiguity exists (e.g., whether a string is a file hash or random hex), the system performs confidence checks via post-processing heuristics or performs an optional validation pass through the LLM for schema correction. Because REX outputs structured artifacts rather than free-form text, it also provides reliable anchor points for dependency tracking and pipeline auditing, enabling consistent behavior across threat hunting tasks from attribution to mitigation.

**Design Rationale:** Regex parsing remains indispensable because many threat indicators—such as IP addresses, hashes, and domains—follow strict syntactic patterns. Blue team analysts often must quickly normalize raw log data or incident feeds into structured formats suitable for correlation across SIEM or TIP platforms. Automated regex-based extraction delivers high precision and avoids false alarms, providing reliable building blocks for pivoting investigations, linking disparate alerts, and enriching threat databases with verified observables.

### B.3. SUM (Summarization)

The Summarization (SUM) module focuses on compressing lengthy, often repetitive cybersecurity narratives into analyst-ready intelligence while preserving operationally critical content. In contrast to general-purpose summarization, SUM is optimized for blue-team workflows and retains domain-specific elements such as TTP identifiers, IOCs, victims, exploit chains, and mitigation references.

The SUM function is also task-aware: depending on the stage in the dependency chain (e.g., behavior analysis vs. response & mitigation), it adjusts its compression ratio and focus. Summaries are stored and propagated forward, e.g., the MAP function uses prior summaries to map indicators to security tools or ATT&CK techniques. Because the module is prompt-based rather than model-specific, it can be upgraded or extended without retraining, and it also supports multi-format summarization (single paragraph, bullet points, timeline, executive brief).

**Implementation.** SUM operates as a controlled LLM inference wrapped in a function call it's applied via a summarization prompt with explicit coverage requirements.

**Prompt Template: Structured Cyber Threat Summarization**

**System Prompt:**
You are a cybersecurity threat intelligence analyst. Your task is to distill the essential technical and investigative details from the following report while preserving operational relevance. The summary will be fed into downstream

analysis modules, so factual precision and coverage are critical.

**Instructions:** Summarize the report into a concise paragraph (3–5 sentences) that includes:

- Attack vector(s) used (e.g., phishing, RCE, supply chain)

- Key TTPs, tools, payloads, or malware families

- Indicators of compromise (IPs, domains, hashes, filenames)

- Affected systems, platforms, or business units

- Timeline elements (dates, sequence of notable events)

- Threat actor or attribution context (if mentioned)

- ...<Exhaustive list omitted>...

**Output Format:**

- Output a single plain-text summary paragraph.

- Do not include reasoning steps or formatting.

- Do not add information not grounded in the text.

- Avoid generic cybersecurity language.

- ...<Exhaustive list omitted>...

**User Input:**
<INSERT REPORT TEXT HERE>

**Fault Tolerance.** We further check whether the final output includes required elements like TTPs or IOCs; if not, a corrective round is triggered automatically using a refinement prompt. This enables analysts to obtain high-value summaries even when reports contain redundant text, marketing language, or narrative filler.

**Design Rationale:** Threat reports and advisories are typically lengthy, verbose, and include redundant or irrelevant details. In time-critical investigations, analysts need condensed yet accurate snapshots that retain attack vectors, key actors, and affected assets. Automated summarization provides blue teams with quick situational awareness, enabling them to brief stakeholders or prioritize triage without missing essential context. It also helps align tactical actions with strategic threat intelligence by stripping away noise and surfacing the essentials.

### B.4. SIM (Text Similarity Matching)

To determine semantic equivalence between pairs of threat indicators, particularly geographic or cultural references (e.g., "Eastern European" vs. "Russian-speaking"), the SIM function applies LLM-based textual similarity matching. This is critical for normalizing contextual descriptions found in incident reports or threat assessments that use varied, informal, or aliasing terms to describe similar threat origin profiles. Rather than relying on surface-level keyword overlap, SIM leverages the LLM's contextual understanding to judge whether two descriptions refer to the same underlying group or region. This helps unify disparate threat intelligence entries that may use different terminology for the same adversarial origin.

**Implementation.** SIM is invoked as a lightweight operation that accepts text pairs and returns a structured similarity judgment. While LLMs perform the reasoning, the module enforces a constrained output schema (boolean + confidence + justification) to support downstream automation. Confidence calibration is handled through a scoring strategy—models may either (1) generate a self-assessed confidence score or (2) be wrapped with a probabilistic calibration layer (e.g., logit normalizers or temperature scaling) to produce stable outputs for downstream classifiers. In chain-of-thought restricted environments, the justification can be generated independently using controlled prompting.

**Prompt Template: Threat Intelligence Similarity Matching**

**System Prompt:**
You are a cybersecurity threat intelligence analyst. Your task is to determine whether two textual descriptions refer to the same underlying entity, region, or concept in a cyber threat context. Focus on meaning, not word overlap.
**Instructions:** Given two text snippets, decide whether they semantically refer to the same threat origin or classification. Consider the following criteria:

- Do both terms refer to the same region, culture, language group, or geopolitical sphere?

- Would threat intelligence analysts commonly treat them as interchangeable or equivalent?

- Does one term logically imply the other (e.g., hierarchical or subset relationship)?

- ...<Exhaustive list omitted>...

**Output Format (JSON):**

```
{
  "match": true/false,
  "confidence": <float from 0.0 to 1.0>,
  "justification": "<one or two sentences>"

}
```

**Rules:**

- Do not hallucinate geopolitical facts not grounded in common CTI usage.

- Avoid vague language; be explicit and concise.

- If uncertain, return "false" with lower confidence.

- ...<Exhaustive list omitted>...

**User Input:** "text-a": "<PHRASE 1>", "text-b": "<PHRASE 2>"

**Design Rationale:** Threat hunting often suffers from inconsistent terminology—analysts and vendors may describe the same adversary group or region in different ways. By applying semantic similarity matching, blue teams can unify aliases, regional descriptions, or contextual cues, thereby avoiding fragmented analysis. For example, detecting that "Eastern European actors" and "Russian-speaking threat groups" likely refer to the same set of adversaries allows more coherent attribution and prevents intelligence silos that adversaries can exploit.

### B.5. MAP (Text Mapping)

To visualize and semantically relate named entities and key concepts extracted from cybersecurity documents, the **MAP** function supports construction of structured representations such as knowledge graphs or threat maps. These representations help uncover infrastructure relationships, campaign patterns, and geotemporal dynamics in threat activity. When powered by large language models, MAP enables flexible and context-aware extraction of relational triples from unstructured threat reports.

**Implementation.** MAP is implemented as a controlled function call that prompts an LLM to convert threat report text into relational triples using a constrained grammar. To ensure consistency, MAP operates on top of the structured output of upstream modules (e.g., NER, REX) and inherits their normalization. When reports are long, the system chunk-summarizes relevant sections before triple extraction. Triples are validated using schema rules and optionally cross-referenced with public ontologies (e.g., MITRE ATT&CK, CAPEC, CVE metadata) to filter impossible or low-confidence edges.

**Prompt Template: Knowledge Graph Triple Extraction for Threat Intelligence**

**System Prompt:**
You are a cybersecurity knowledge mapping assistant. Your task is to convert the threat report into structured triples suitable for knowledge graph construction. Focus on relationships that matter for attribution, behavior analysis, and mitigation planning.

**Instructions:** From the text below, extract subject–predicate–object triples, using only information grounded in the text. Examples of valid predicates include (but are not limited to):

- **uses / deploys / distributes** (actor → malware/tool)

- **exploits / targets** (tool or actor → vulnerability or victim)

- **communicates with / hosts / controls** (tool → infrastructure or C2)

- **linked to / attributed to / associated with** (malware or campaign → threat actor)

- **observed in / active since** (indicator → date or region)

**Output Format (JSON Array):**

```
[
  {
    "subject": "...",
    "predicate": "...",
    "object": "...",
    "confidence": <float between 0.0 and 1.0>
  },
  ...
]
```

**Rules:**

- Triples must be grounded in the text — no speculation or external facts.

- Keep entities normalized (e.g., uppercase CVEs, lowercase domains).

- Avoid vague predicates like "related to" if a more precise one applies.

- Use a new entry for each distinct triple — do not chain multiple relations into one object.

- If uncertain, include the triple with a low confidence value rather than omitting it.

**User Input:** <INSERT REPORT TEXT HERE>

**Design Rationale:** Attack campaigns rarely consist of isolated events—they are orchestrated through complex infrastructures and actor-tool relationships. Mapping extracted entities into structured knowledge graphs helps analysts visualize these relationships and trace adversary activity across time and geography. This capability supports detection of infrastructure reuse, identification of campaign evolution, and discovery of hidden connections that might otherwise remain unnoticed, enabling more proactive defense strategies and long-term threat tracking.

### B.6. RAG (Retrieval-Augmented Generation)

To enhance generation with accurate and recent data, RAG combines LLM output with real-time retrieval from external threat intelligence APIs or databases. It is particularly useful for describing evolving threats or identifying actor affiliations.

**Implementation.** The RAG module is implemented as a hybrid retrieval system that combines a local, index-backed threat intelligence store with real-time access to external APIs.

Case 1: To support fast and semantically relevant lookup, we construct a vector database using FAISS as the primary indexing backend. Incoming documents including CVE entries, MITRE ATT&CK profiles, malware analyses from threat

intelligence vendors, advisories from CISA, and campaign reports from industry sources (exhaustive list in Appendix A) are first normalized and split into passages. Each passage is embedded using a domain-optimized sentence transformer, and both embeddings and metadata (source, timestamp, threat type, confidence flags) are stored for retrieval. A lightweight SQLite or Elasticsearch layer stores metadata alongside embeddings, enabling structured filtering based on source authority, document recency, and entity type. This indexing strategy allows RAG to efficiently retrieve highly relevant evidence even when dealing with hundreds of thousands of threat intelligence records.

Case 2: Beyond local indexing, the module supports external retrieval for evolving threats that may not yet exist in the offline database. To do this, the LLM generates a structured search query, which is passed to external providers such as VirusTotal, OTX, Shodan, Recorded Future, or even search engines scoped with domain restrictions (e.g., site:mitre.org or site:cisa.gov). Retrieved results are normalized and converted into candidate passages, which are embedded and re-ranked against internal results using a hybrid scoring function combining dense similarity, BM25 keyword matching, and optional LLM-based relevance scoring. If external sources fail to respond or return low-quality results, the system automatically falls back to the local vector index, maintaining robustness and availability during inference.

The final generation process is executed by two steps. First, the system returns a retrieval context composed of top-ranked passages and metadata. Then, the LLM is prompted to synthesize an answer grounded strictly in retrieved evidence. To prevent hallucination, citations are enforced through schema validation, and responses are rejected if they contain assertions unsupported by retrieval. The module also logs all retrieved evidence into the shared working memory layer, enabling downstream modules such as MAP for threat mapping or SUM for report summarization to reuse the same evidence without redundant retrieval. By combining local indexing with external live search and pairing retrieval with generation under strict grounding constraints, the RAG module supports high-fidelity, up-to-date analysis while preserving transparency and auditability in blue-team threat hunting.

**Design Rationale:** Adversary tactics evolve daily, and static LLMs quickly become outdated if disconnected from real-time sources. Retrieval-augmented generation enables blue teams to ground LLM outputs with fresh, authoritative information from trusted CTI feeds, vulnerability databases, or public repositories. This ensures that generated insights remain both accurate and timely, supporting decisions during live incidents such as phishing outbreaks or zero-day exploitation campaigns where stale intelligence could lead to ineffective responses.

### B.7. SPA (Text Span Localization)

To precisely extract actionable phrases (e.g., indicators of compromise or technique descriptions) from long-form cybersecurity text, **Text Span Localization** (SPA) models are used.

**Implementation.** SPA first uses semantic search to narrow down text windows containing relevant context, then applies a prompt-based extraction pass where the LLM returns the minimal span satisfying the query. For improved precision, the extracted span is validated using token-based alignment heuristics that measure similarity against known technique glossaries (e.g., MITRE ATT&CK descriptions). If the output is too vague, over-extended, or incomplete, a refinement loop is triggered to reissue the prompt with tightened constraints. Outputs are scored using two complementary metrics: Exact Match (EM) for strict correctness, and Intersection-over-Union (IoU) to measure partial overlap between predicted and ground-truth spans. Detailed formulations are provided below:

- **Exact Match (EM)**:
$$\text{EM} = \frac{\text{Number of exact matches}}{\text{Total predictions}}$$

- **Intersection over Union (IoU)**:
$$\text{IoU} = \frac{|S_p \cap S_t|}{|S_p \cup S_t|}$$

---

**Prompt Template: Targeted Span Extraction for Cyber Threat Reports**

**System Prompt:**
You are a cybersecurity span localization assistant. Your goal is to extract the minimal text span that directly describes the attack technique used in the incident. The output must match the source text exactly, not a paraphrase.

**Evaluation Criteria (Built Into the Task):** Your extracted span will be evaluated using two metrics:

- **Exact Match (EM):** You get full credit only if your extracted text exactly matches the ground truth span with no extra or missing tokens.

- **Intersection over Union (IoU):** Partial credit is awarded based on the overlap between your extracted span $S_p$ and the true span $S_t$:

$$\text{IoU}(S_p, S_t) = \frac{|S_p \cap S_t|}{|S_p \cup S_t|}$$

To maximize both scores, return the shortest possible span that fully captures the technique description without adding unrelated text.

**Instructions:** From the text below, extract the exact sentence or phrase that describes the primary attack technique or method of compromise (e.g., spearphishing, RCE, credential dumping, lateral movement). The output must be a direct substring of the input.

**Output Format:**

```
"<EXACT_SPAN_FROM_TEXT>"
```

**Rules:**

- Return only the span — no commentary or explanation.

- The extracted text must be contiguous and appear exactly in the input.

- If multiple spans qualify, return the most complete or specific one.

- If no valid span is present, return an empty string.

**User Input:** <INSERT REPORT EXCERPT HERE>

**Design Rationale:** In practice, analysts often need to pull out the single critical phrase—such as the exact exploitation method—from long reports or alerts. Span localization ensures precision by targeting actionable fragments rather than broad summaries, which is vital for creating detection rules, YARA signatures, or SIEM correlation logic. By pinpointing exact techniques or IOCs, blue teams reduce ambiguity, streamline evidence curation, and avoid wasting resources on imprecise or overly generalized intelligence.

### B.8. CLS (Classification)

To measure the ability of a system to categorize cybersecurity-relevant textual inputs into predefined classes (e.g., threat categories, severity levels, or attack types), classification models are employed.

**Implementation.** We implement CLS using transformer-based large language models (LLMs), which utilize a special token (e.g., [CLS]) to represent sentence-level semantics. The resulting embedding is mapped to labels through a learned classifier.

**Design Rationale:** Blue teams constantly receive heterogeneous data ranging from phishing alerts to vulnerability disclosures. Automated classification allows this information to be triaged into relevant categories (e.g., attack type, severity, or impacted systems) so that workflows can be routed efficiently. Accurate classification supports prioritization of critical alerts, ensures compliance with response playbooks, and minimizes analyst fatigue by filtering out low-severity noise while surfacing the incidents that require immediate attention.

### B.9. MATH (Mathematical Calculation)

To perform quantitative analyses and structured computations relevant to cybersecurity, the **MATH** function supports tasks such as frequency modeling, impact scoring, cryptographic evaluation, and automated threat prioritization. These computations are critical for risk-informed decision-making within cyber threat intelligence pipelines.

**Implementation.** Considering that MATH is mainly used for severity quantification, we thus implement it through **Common Vulnerability Scoring System (CVSS v3.1)**, which uses a combination of weighted factors and conditional logic to produce

a standardized severity score for vulnerabilities. One key element is the *Base Score*, calculated using the Impact and Exploitability sub scores:

$$\text{Base Score} = \begin{cases} 0, & \text{if Impact Subscore } \leq 0 \\ \text{RoundUp}\left(\min(\text{Impact} + \text{Exploitability}, 10)\right), & \text{if Scope is Unchanged} \\ \text{RoundUp}\left(\min(1.08 \times (\text{Impact} + \text{Exploitability}), 10)\right), & \text{if Scope is Changed} \end{cases}$$

The *Impact Subscore* is computed from confidentiality, integrity, and availability impact metrics as:

$$\text{ISC}_{\text{Base}} = 1 - (1 - C) \times (1 - I) \times (1 - A)$$

This formula models the probability that the system's security properties are affected by a vulnerability. The resulting score guides patching priority, risk exposure assessments, and automated vulnerability triage.

Such logic-heavy, non-trivial calculations exemplify the role of mathematical modules in operational cybersecurity settings and justify the integration of computational reasoning capabilities in modern cyber AI systems.

**Design Rationale:** Quantitative scoring frameworks like CVSS remain the backbone of enterprise vulnerability management and patch prioritization. Automated mathematical reasoning allows blue teams to consistently compute, validate, and apply these scores across large vulnerability sets, ensuring consistent triage even under heavy load. Beyond CVSS, mathematical modules enable probability modeling, risk scoring, and exposure forecasting—practices that help defenders allocate resources effectively and justify decisions to leadership with evidence-based metrics.

## C. Metric

Below are further details on how each evaluation metric quantifies the corresponding threat hunting performance.

### C.1. Generation (Precision–Recall Balance by F1) and Classification (Accuracy)

In threat hunting, information extraction tasks such as detecting malware names, extracting IOCs, or identifying exploited vulnerabilities require a careful balance between precision and recall. If a system retrieves too many irrelevant indicators, analysts are burdened with noise; if it misses critical signals, adversarial activity may go unnoticed. The **F1 score** captures this balance by evaluating how well a model retrieves the right items while minimizing both false alarms and missed detections. This makes it particularly valuable in operational contexts where the completeness and reliability of extracted intelligence directly affect the quality of subsequent analysis and response.

Besides, well-quantified tasks such as prioritization in blue team activities involve classification, such as determining whether an alert corresponds to privilege escalation, categorizing attack vectors, or assigning severity levels to vulnerability reports. In these scenarios, **accuracy** serves as an intuitive and effective measure of system performance, reflecting how often predictions align with ground-truth categories. High accuracy ensures that automated classification supports efficient triage and aligns with established taxonomies like MITRE ATT&CK.

### C.2. Sim (BERT Score)

To evaluate the semantic similarity between cybersecurity-related texts—such as comparing analyst-written threat summaries, aligning generated incident narratives with original reports, or verifying paraphrased explanations of threat indicators—the **Sim** function utilizes contextual embedding-based metrics. Specifically, it computes **BERTScore** (Zhang et al., 2020), which has been shown to correlate strongly with human judgment in natural language generation tasks.

BERTScore measures semantic equivalence at the token level by aligning contextual embeddings from pre-trained transformer models. The score is computed as:

$$\text{BERTScore} = \frac{1}{|x|} \sum_{i} \max_{j} \cos(\mathbf{x_i}, \mathbf{y_j})$$

where $\mathbf{x_i}$ and $\mathbf{y_j}$ are contextual embeddings of tokens in the candidate and reference texts, respectively. The final score reflects the average of maximal cosine similarities for each token in the candidate sentence.

This metric is particularly valuable in evaluating machine-generated text in cybersecurity domains, where surface-level similarity may fail to capture the deeper equivalence of technical meaning or threat context.

### C.3. Pass (Code Execution Passing Rate)

To measure the reliability and functional correctness of cybersecurity automation artifacts—such as detection rules, analysis scripts, or integration workflows—the **Pass Rate** metric is employed. It quantifies how well a system performs under test by evaluating the proportion of test cases that execute successfully within a defined execution cycle, often conducted in a continuous integration (CI) pipeline.

Formally, the Pass Rate is defined as:

$$\text{Pass Rate} = \frac{\text{Number of Passed Tests}}{\text{Total Tests Executed}} \times 100\%$$

This metric provides a coarse yet effective indicator of operational readiness. A high Pass Rate implies that the deployed codebase functions as intended across its tested scenarios, which is critical in cybersecurity contexts where automation is used to process threat intelligence, detect anomalies, or trigger incident response mechanisms.

Routine monitoring of this metric supports the early identification of integration regressions, promotes pipeline stability, and ensures confidence in deploying automated defensive measures to production environments.

### C.4. Hit (Top-k Hit Ratio)

To evaluate the effectiveness of cybersecurity recommendation or retrieval systems—such as those that propose relevant threat indicators, patch suggestions, attack techniques, or investigative leads—the **Top-k Hit Ratio** is employed. This metric measures how frequently at least one correct or relevant item appears within the top-$k$ ranked results returned by the system.

Mathematically, the Top-k Hit Ratio is defined as:

$$\text{Hit@}k = \frac{\text{Number of queries with at least one relevant item in top } k}{\text{Total number of queries}}$$

A higher Hit@k indicates better system performance in surfacing relevant intelligence near the top of recommendations, which is critical for time-sensitive security operations.

**Use Case Example:** If a system recommends threat indicators based on a query about a ransomware family, Hit@5 evaluates whether at least one valid IOC (e.g., file hash or C2 domain) appears in the top 5 returned items.

---

**Prompt 6. Hit Evaluation Prompt for Threat Retrieval**

**System Prompt:** You are an assistant for evaluating cybersecurity retrieval systems. Given a query and a list of system-generated recommendations, check whether any ground truth item appears within the top-$k$ returned results.
**Instructions:** For each query, compare the top-$k$ predicted items against the gold-standard set. Indicate `"Hit"` if at least one match exists, otherwise `"Miss"`.
**Output:** Return a JSON object with fields: `query`, `top_k_results`, `ground_truth`, `hit@k`: true/false

---

### C.5. Dist (Normalized Distance Similarity)

To evaluate the accuracy of numeric predictions in range-based estimation tasks, such as severity scoring, the **Normalized Distance Similarity** (**Dist**) metric is employed. This metric compares the predicted number and the ground-truth and scales the similarity into the $[0, 1]$ range, where higher values indicate closer alignment.

Formally, the similarity is computed as:

$$\text{Similarity} = 1 - \frac{|\hat{c} - c|}{R}$$

where $\hat{c}$ and $c$ denote the midpoints of the predicted and true ranges, respectively, and $R$ is the maximum possible value of the range (e.g., 10 in our case of CVSS scores). The metric reflects the Euclidean distance between prediction and truth, normalized such that a perfect match yields a similarity of 1, and the furthest possible discrepancy yields 0.

## D. Experimental Setting

This section details the experimental setup used to evaluate LLMs in the CyberTeam benchmark.

Hyperparameters. Table 4 summarizes the key hyperparameters for querying LLMs during experiments. These settings were chosen to balance generation quality and computational efficiency.

*Table 4.* LLM query hyperparameters.

| Hyperparameter | Value | Description |
|---|---|---|
| Temperature | 0.7 | Output randomness |
| Top-p | 0.95 | Nucleus sampling threshold |
| Max tokens | 2048 | Generation length cap |
| Stop sequences | `["\n", "Q:"]` | Response cutoff cues |
| Prompt format | ICL, CoT, ToT, Emb | Prompt types (see 4) |
| Tool-calling API | Enabled (Selective) | For function-use experiments |

**Computational Resources.** All experiments were conducted on a high-performance computing cluster equipped with six NVIDIA RTX 6000 Ada Generation GPUs, each with 48 GB of dedicated VRAM. The system utilized CUDA version 12.8 and NVIDIA driver version 570.124.06. This configuration enabled parallel execution of model inference, evaluation, and tool-augmented tasks across the benchmark datasets. The hardware provided sufficient memory bandwidth and processing power to handle large-scale experiments, including multi-sample prompting strategies like CoT and ToT, without encountering resource constraints. Each experimental run was executed in a isolated environment to ensure reproducibility and avoid interference between tasks.

## E. Additional Experimental Results

This section presents additional experimental results that complement our main findings, offering deeper insights into model behavior across varied threat-hunting scenarios.

### E.1. Running Time and Trade-off between Latency and Effectiveness

**Observations and Insights.** The runtime analysis highlights an inherent trade-off between efficiency and reasoning complexity across prompting strategies. Consistent with expectations, in-context learning (ICL) variants remain the fastest across nearly all models, typically completing tasks in the 10–15 second range. This makes ICL attractive for time-sensitive operations such as triage or initial correlation, where speed outweighs the need for more structured reasoning. Chain-of-thought (CoT) introduces additional reasoning overhead, increasing runtimes by roughly 30–40% compared to ICL. While this slowdown is measurable, the benefit of CoT lies in its improved consistency on more nuanced decision tasks, suggesting that blue teams might selectively invoke CoT when precision is critical. Tree-of-thought (ToT), by contrast, incurs the highest latency, often doubling the runtime relative to ICL. This stems from ToT's multi-branch exploration process, which, while occasionally producing richer reasoning chains, remains computationally expensive and operationally impractical for most real-time security workflows.

Our standardized pipeline approach falls between CoT and ToT in runtime. The added latency reflects the sequential decomposition of tasks into modular subroutines, each enforcing more structured reasoning than raw prompting. While slower than single-pass approaches, **our pipeline mostly avoids the extreme overhead observed in ToT.** This stability is particularly important in operational settings: analysts can predictably plan around a known latency budget while still benefiting from higher reliability and repeatability of results.

*Table 5.* Running time (in seconds) of LLMs on CYBERTEAM, comparing different open-ended prompting strategies with our standardized method. Lower values indicate faster inference.

| Method | | Cybersecurity Agent | | | Industry-Leading LLM | | | | | | | |
|---|---|---|---|---|---|---|---|---|---|---|---|---|
| | | LY | DH | SL | GK | G5 | QW | GM | CD | L3.1 | L4 | GA |
| *Playbook Recommend* | | | | | | | | | | | | |
| | ICL5 | 12.4 | 15.6 | 14.8 | 10.5 | 41.2 | 13.6 | 11.9 | 12.7 | 28.6 | 24.0 | 16.8 |
| | ICL10 | 14.1 | 17.2 | 16.3 | 12.0 | 45.7 | 15.2 | 13.5 | 14.4 | 31.4 | 26.5 | 18.3 |
| Open-ended | CoT | 18.6 | 22.4 | 21.1 | 15.8 | 60.8 | 19.9 | 17.6 | 18.8 | 41.2 | 34.5 | 24.5 |
| | ToT | 27.5 | 34.1 | 32.0 | 24.2 | 89.5 | 30.2 | 27.1 | 28.9 | 62.7 | 52.5 | 37.8 |
| **Standardized (Ours)** | | 21.3 | 26.5 | 25.2 | 19.1 | 71.3 | 23.5 | 21.0 | 22.3 | 50.5 | 42.0 | 30.1 |
| *Security Control Adjust* | | | | | | | | | | | | |
| | ICL5 | 13.1 | 16.4 | 15.5 | 11.1 | 43.5 | 14.3 | 12.5 | 13.3 | 30.2 | 25.0 | 17.6 |
| | ICL10 | 15.0 | 18.3 | 17.2 | 12.6 | 48.1 | 16.1 | 14.2 | 15.1 | 33.0 | 27.4 | 19.1 |
| Open-ended | CoT | 19.4 | 23.5 | 22.2 | 16.7 | 63.4 | 20.8 | 18.3 | 19.6 | 43.8 | 36.0 | 25.7 |
| | ToT | 28.3 | 35.6 | 33.6 | 25.4 | 92.2 | 31.7 | 28.3 | 30.2 | 66.4 | 55.0 | 39.5 |
| **Standardized (Ours)** | | 22.1 | 27.8 | 26.7 | 20.0 | 74.2 | 24.7 | 22.1 | 23.4 | 53.1 | 44.0 | 31.2 |
| *Patch Code Generation* | | | | | | | | | | | | |
| | ICL5 | 14.2 | 17.9 | 17.0 | 12.2 | 46.8 | 15.7 | 13.6 | 14.4 | 32.8 | 27.0 | 19.2 |
| | ICL10 | 16.3 | 19.6 | 18.7 | 13.7 | 51.3 | 17.5 | 15.3 | 16.2 | 35.7 | 29.5 | 20.8 |
| Open-ended | CoT | 21.2 | 25.3 | 24.5 | 18.1 | 68.7 | 22.9 | 19.7 | 21.1 | 47.6 | 39.0 | 28.1 |
| | ToT | 31.7 | 38.4 | 36.9 | 27.8 | 98.6 | 34.5 | 30.6 | 32.6 | 71.9 | 59.0 | 42.5 |
| **Standardized (Ours)** | | 24.0 | 29.7 | 28.6 | 21.4 | 79.4 | 26.6 | 23.6 | 25.0 | 57.2 | 47.0 | 34.4 |
| *Patch Tool Suggestion* | | | | | | | | | | | | |
| | ICL5 | 12.8 | 15.9 | 15.2 | 10.9 | 42.6 | 14.0 | 12.3 | 13.0 | 29.5 | 24.3 | 17.0 |
| | ICL10 | 14.7 | 17.7 | 16.9 | 12.4 | 47.0 | 15.8 | 14.0 | 14.8 | 32.4 | 26.2 | 18.6 |
| Open-ended | CoT | 19.0 | 23.0 | 22.0 | 16.3 | 62.1 | 20.5 | 18.1 | 19.2 | 42.5 | 35.0 | 25.1 |
| | ToT | 27.9 | 34.9 | 33.1 | 24.7 | 90.8 | 31.0 | 27.6 | 29.6 | 64.0 | 52.5 | 38.2 |
| **Standardized (Ours)** | | 21.7 | 27.1 | 26.2 | 19.6 | 72.8 | 24.1 | 21.7 | 22.9 | 51.7 | 42.5 | 30.5 |
| *Advisory Correlation* | | | | | | | | | | | | |
| | ICL5 | 13.6 | 16.8 | 16.1 | 11.7 | 44.9 | 14.9 | 13.0 | 13.8 | 31.0 | 25.5 | 18.2 |
| | ICL10 | 15.6 | 18.7 | 17.9 | 13.2 | 49.6 | 16.7 | 14.7 | 15.6 | 34.0 | 28.0 | 20.0 |
| Open-ended | CoT | 20.3 | 24.1 | 23.4 | 17.2 | 65.3 | 21.7 | 19.0 | 20.4 | 45.3 | 37.0 | 26.5 |
| | ToT | 29.8 | 36.8 | 35.4 | 26.1 | 95.1 | 33.1 | 29.4 | 31.3 | 68.8 | 56.0 | 40.1 |
| **Standardized (Ours)** | | 23.1 | 28.5 | 27.8 | 20.6 | 76.2 | 25.4 | 22.8 | 24.2 | 54.6 | 45.0 | 32.1 |

Unlike open-ended reasoning, which may fluctuate in quality depending on the model and prompt, the standardized pipeline enforces uniform logic steps, reducing error propagation at the cost of additional inference time. From a deployment standpoint, this balance offers a pragmatic middle ground: not as lightweight as ICL for quick heuristics, but substantially more usable than ToT when analysts demand repeatable outputs.

### E.2. Additional Results of Individual Threat Hunting Performance

Figure 5, 6, 7, and 8 complement the results as present in Figure 3, offering aligned insights as exhibited in previous experiments.

Based on those results, we further outline the following observations and analyses:

**Attribution-Oriented Tasks.** Attribution tasks rely on aligning disparate indicators into coherent profiles of adversaries, infrastructure, and campaigns. Here, the standardized workflow shows its greatest benefit because it forces the model to treat each extracted clue as part of a larger dependency chain. When the reasoning is left open ended, models often generate fluent narratives that omit critical ties, such as overlooking how infrastructure relates to a specific campaign or how victimology patterns reinforce an actor hypothesis. The modular approach ensures that entity recognition, context mapping, and relational inference are explicitly sequenced, which reduces the tendency of the model to drift or collapse multiple

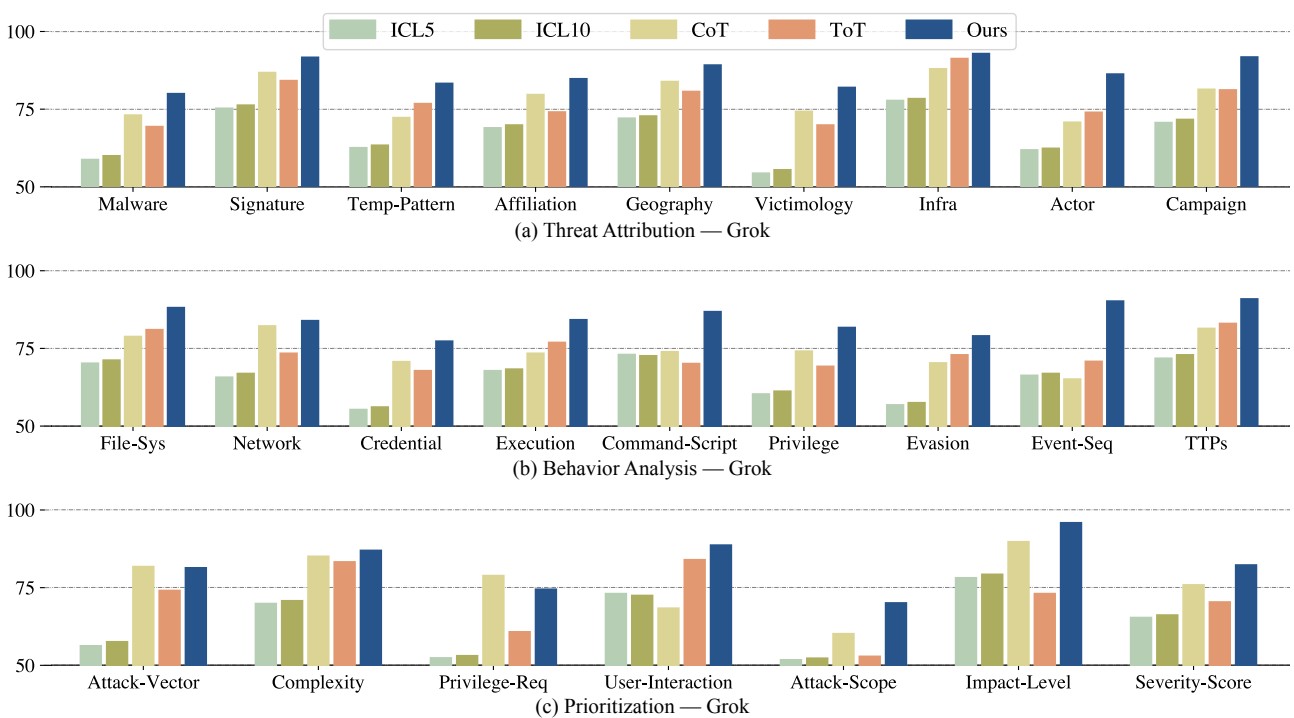

*Figure 5.* Threat-hunting performance on individual tasks, evaluating under Grok.

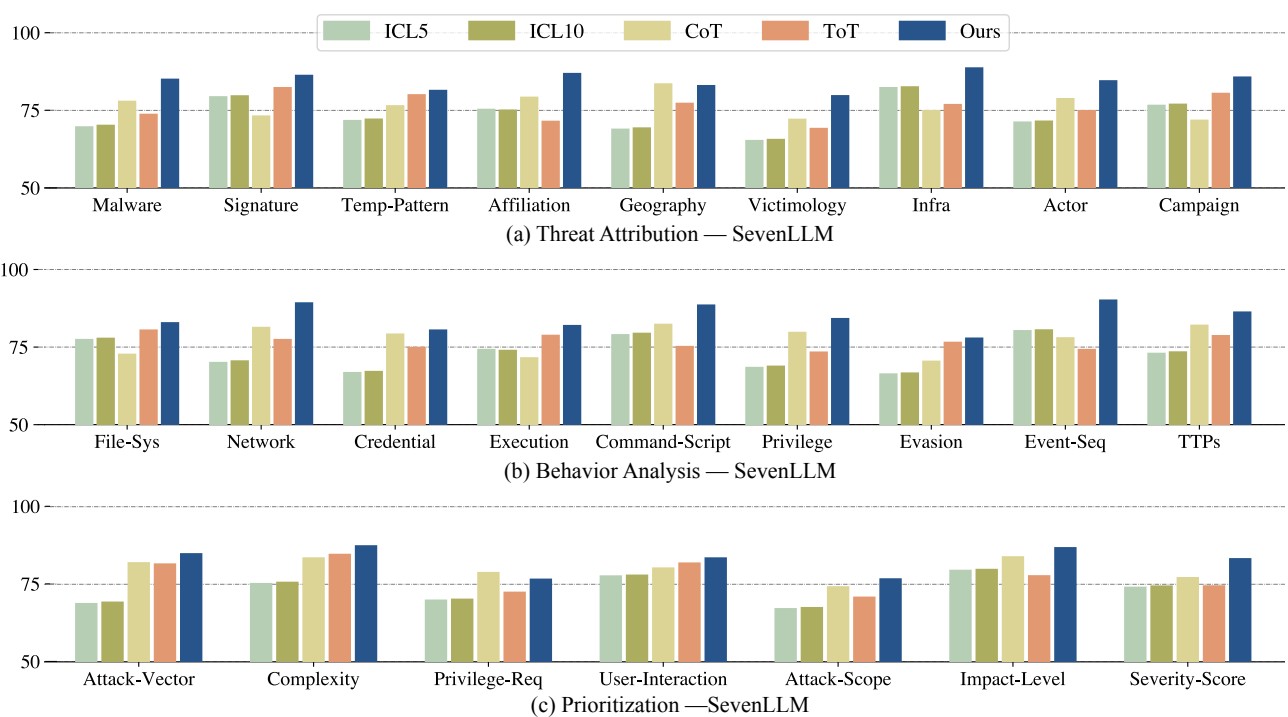

*Figure 6.* Threat-hunting performance on individual tasks, evaluating under SevenLLM-7B.

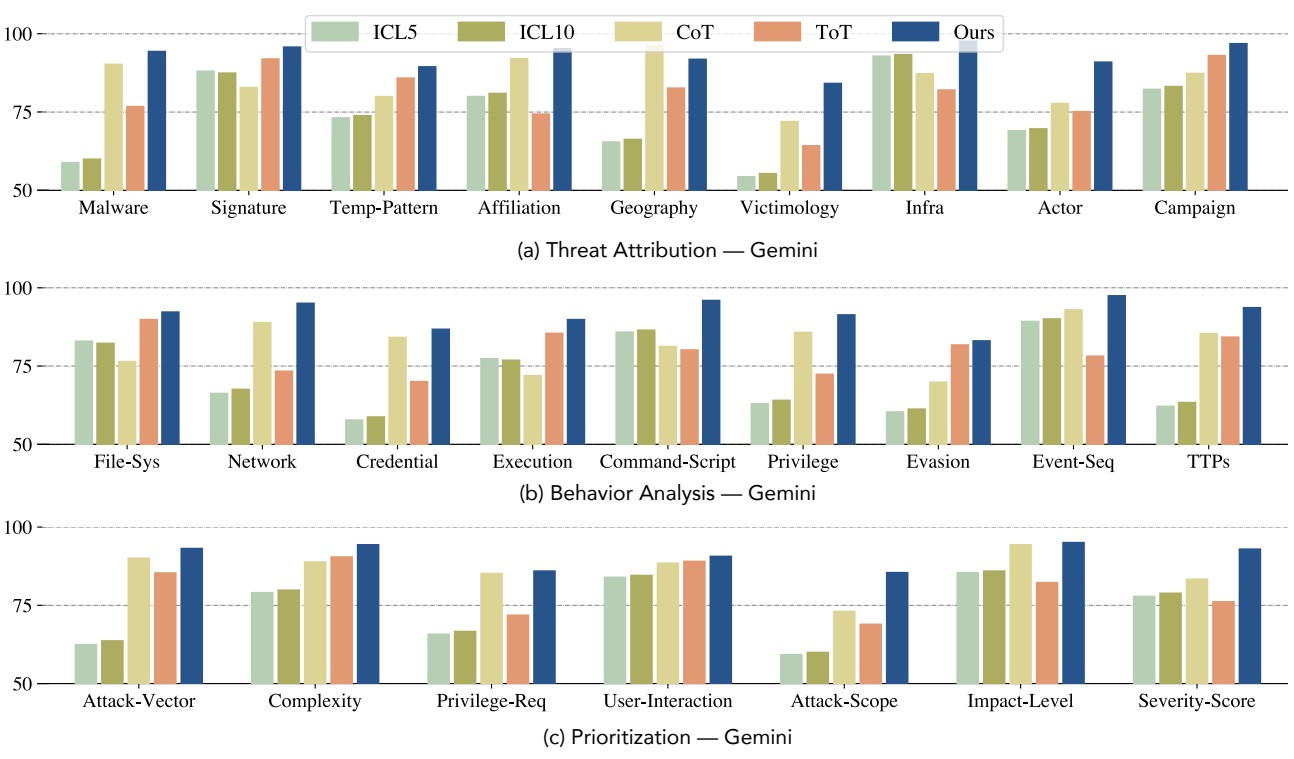

*Figure 7.* Threat-hunting performance on individual tasks, evaluating under Gemini-pro.

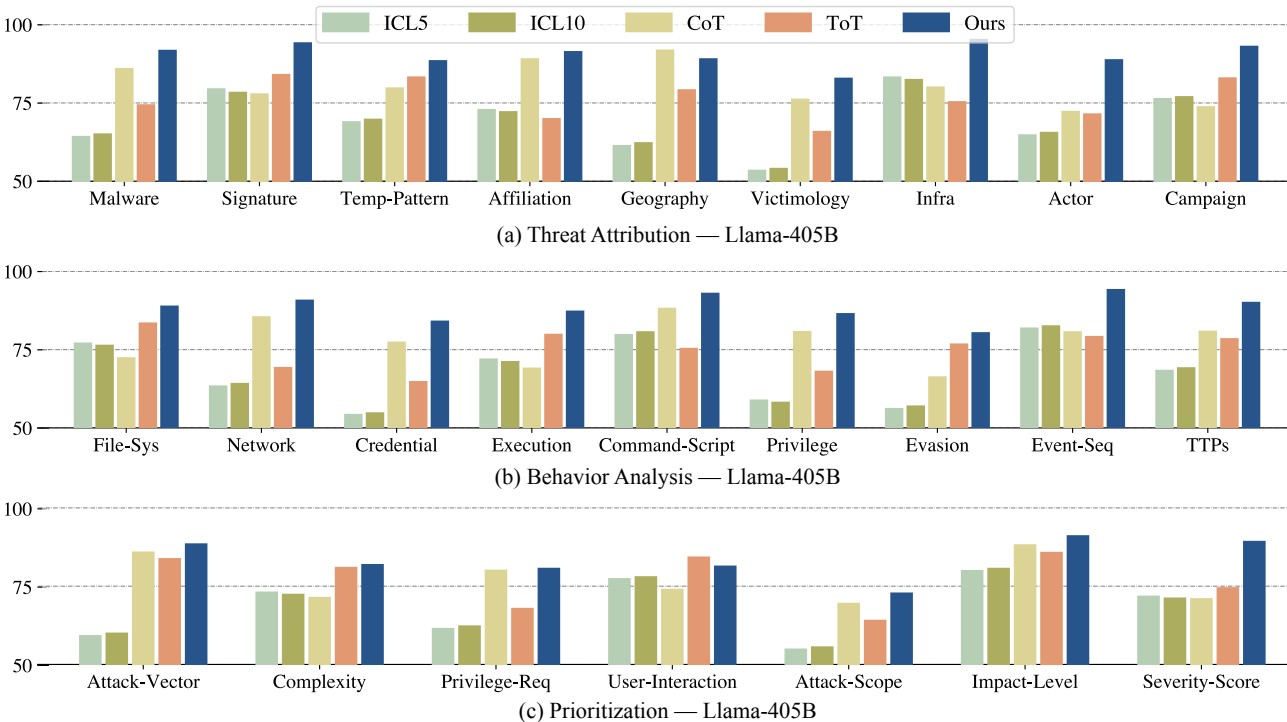

*Figure 8.* Threat-hunting performance on individual tasks, evaluating under Llama-405B.

actors into a generic label. This structured pipeline also helps the model preserve continuity across steps, so that information about geography, malware signatures, or campaign overlaps is not forgotten or misapplied. In practice, this makes attribution outputs more consistent and trustworthy, with fewer contradictions across different facets of the same incident.

**Behavioral-Oriented Tasks.** Behavioral analysis tasks focus on describing how an attack unfolds across file systems, networks, credentials, execution flows, and evasion strategies. These tasks expose a different challenge: models must reason not just about isolated labels but about temporal or causal sequences. Open ended reasoning often struggles to maintain logical order, for example by misplacing the relationship between a credential theft and subsequent privilege escalation, or by skipping intermediate steps in an event sequence. Standardized workflows address this by explicitly guiding the model to construct event chains step by step, preserving both order and dependency. This guidance is particularly important when behaviors are nested, such as when command script execution spawns further lateral movements or when evasion strategies are intertwined with persistence mechanisms. The modular design ensures that contextual cues are not discarded midway, producing outputs that resemble the structured analysis human analysts expect. The gains here are not simply about accuracy but about interpretability, as the resulting narratives make it easier to understand how behaviors connect to form a complete attack path.

**Prioritization-Oriented Tasks.** Prioritization tasks require models to map extracted observations into judgments of impact, severity, or scope. These are less about narrative flow and more about logical consistency and rule following. While open ended reasoning can handle straightforward labels like user interaction requirements or attack complexity, it often falters when multiple inputs must be integrated into a composite assessment. For example, determining severity requires careful alignment of impact level, attack vector, and privilege requirements, which is difficult to achieve reliably without structured steps. The standardized workflow enforces this alignment by ensuring that each component assessment is produced systematically and then fed into the final prioritization judgment. As a result, the model is less likely to generate inconsistent or contradictory scores. The benefits are particularly visible in tasks that resemble rule based calculations or scoring rubrics, where the modular structure mirrors the procedural way that human analysts reason about risk.

**Broader Implications.** Viewed across these categories, a clear pattern emerges. Attribution tasks benefit most from the preservation of contextual dependencies across different indicators. Behavioral tasks gain from the ability to model temporal and causal structure. Prioritization tasks see improvements in logical consistency and integration of multiple criteria. Standardization does not change the core language modeling capability of these systems, but it channels their generative power into workflows that mirror how analysts actually think about security problems. This alignment between workflow design and task demands is the primary driver of the gains observed, and it demonstrates why modular guidance is most valuable when reasoning requires structured coordination across multiple dimensions.

**Benchmark Generalizability.** Although CyberTeam integrates data from 23 diverse threat intelligence sources, the benchmark is inherently constrained by the selected datasets and threat scenarios. For example, it may not fully represent emerging attack vectors, such as AI-powered phishing or supply chain compromises. Expanding the benchmark to include more recent and varied threat data, as well as cross-domain applications (e.g., IoT or cloud security), would enhance its utility for evaluating LLM generalization in broader cybersecurity contexts.

### E.3. Human Evaluation

In human evaluation, we recruited three security analysts to assess LLM-generated outputs across four stages of the threat-hunting workflow: Threat Attribution, Behavior Analysis, Prioritization, and Response & Mitigation. Evaluators are asked to make binary accept/reject decisions to reflect real-world triage settings, and we report acceptance rates with inter-annotator variance to capture the uncertainty.

The results show that stronger general-purpose LLMs consistently achieve higher acceptance rates, particularly when guided by the standardized reasoning workflow, while smaller or domain-specialized models exhibit greater variance and reduced reliability. Tasks that simply require structured reasoning or multi-step synthesis (e.g., campaign correlation, event reconstruction, and severity scoring) result in higher disagreement among evaluators, whereas modular tasks like IOC extraction or playbook recommendation benefit more from standardized workflows. Overall, the findings support our claim that CYBERTEAM 's modular framework improves interpretability and operational alignment, but also reveal that human oversight remains essential, especially for complex threat hunting scenarios.

*Table 6.* Human evaluation results on the four threat-hunting categories (scaled to 100%). Each cell reports mean acceptance rate $\pm$ standard deviation across 3 human evaluators.

| Task / Subtask | G5 | QW | GM | L4 | LY |
|---|---|---|---|---|---|
| **Threat Attribution** | | | | | |
| Malware Identification | $82.1_{\pm 0.9}$ | $78.5_{\pm 2.7}$ | $72.6_{\pm 1.3}$ | $60.3_{\pm 2.4}$ | $52.9_{\pm 11.4}$ |
| Infrastructure Extraction | $92.4_{\pm 0.5}$ | $87.1_{\pm 4.9}$ | $90.8_{\pm 3.3}$ | $84.6_{\pm 2.1}$ | $65.4_{\pm 3.1}$ |
| Actor Identification | $90.2_{\pm 1.7}$ | $76.8_{\pm 3.1}$ | $93.9_{\pm 1.2}$ | $78.4_{\pm 2.3}$ | $75.8_{\pm 7.2}$ |
| Campaign Correlation | $83.5_{\pm 6.2}$ | $74.2_{\pm 3.4}$ | $89.6_{\pm 1.6}$ | $75.1_{\pm 2.9}$ | $61.3_{\pm 4.0}$ |
| **Behavior Analysis** | | | | | |
| File System Activity Detection | $81.3_{\pm 7.2}$ | $75.4_{\pm 2.8}$ | $89.6_{\pm 1.7}$ | $77.8_{\pm 2.2}$ | $73.5_{\pm 3.1}$ |
| Credential Access Detection | $75.7_{\pm 6.3}$ | $73.4_{\pm 4.3}$ | $63.4_{\pm 2.8}$ | $67.6_{\pm 3.3}$ | $59.3_{\pm 4.5}$ |
| Command & Script Analysis | $82.2_{\pm 2.2}$ | $71.9_{\pm 3.4}$ | $84.6_{\pm 2.1}$ | $72.4_{\pm 3.5}$ | $40.9_{\pm 3.9}$ |
| Event Sequence Reconstruction | $73.6_{\pm 2.6}$ | $65.1_{\pm 3.7}$ | $73.2_{\pm 2.4}$ | $70.3_{\pm 3.8}$ | $53.2_{\pm 10.2}$ |
| **Prioritization** | | | | | |
| Attack Complexity Summarization | $89.2_{\pm 1.7}$ | $67.3_{\pm 3.3}$ | $84.1_{\pm 2.2}$ | $71.1_{\pm 2.9}$ | $63.8_{\pm 4.1}$ |
| Privileges Requirement Detection | $91.7_{\pm 4.9}$ | $72.6_{\pm 3.2}$ | $87.4_{\pm 1.5}$ | $74.1_{\pm 3.0}$ | $70.8_{\pm 3.6}$ |
| Attack Scope Detection | $86.4_{\pm 2.1}$ | $64.9_{\pm 3.8}$ | $81.2_{\pm 2.4}$ | $68.4_{\pm 3.1}$ | $61.9_{\pm 4.0}$ |
| Severity Scoring | $87.8_{\pm 1.9}$ | $66.0_{\pm 3.6}$ | $82.9_{\pm 2.5}$ | $69.7_{\pm 3.4}$ | $62.5_{\pm 4.3}$ |
| **Response & Mitigation** | | | | | |
| Playbook Recommendation | $93.4_{\pm 0.8}$ | $78.1_{\pm 2.9}$ | $88.4_{\pm 1.5}$ | $76.3_{\pm 2.6}$ | $79.2_{\pm 3.0}$ |
| Patch Tool Suggestion | $84.6_{\pm 2.8}$ | $61.2_{\pm 4.2}$ | $78.5_{\pm 2.6}$ | $66.9_{\pm 3.6}$ | $71.4_{\pm 4.7}$ |
| Advisory Correlation | $91.9_{\pm 1.1}$ | $77.8_{\pm 2.7}$ | $87.6_{\pm 1.4}$ | $74.2_{\pm 3.1}$ | $77.5_{\pm 3.3}$ |

*Table 7.* Ablation study results of LLM threat-hunting performance (scaled to 100%).

| Task | Ablation | G5 | QW | GM | L4 | LY |
|---|---|---|---|---|---|---|
| Playbook Recommend | w/o SPA | 88.9 | 76.5 | 89.7 | 77.2 | 65.1 |
| | w/o SUM | 70.3 | 58.7 | 71.4 | 62.9 | 44.2 |
| | w/o RAG | 61.9 | 53.1 | 68.9 | 57.8 | 46.0 |
| | w/o NER | 73.4 | 61.2 | 75.8 | 66.3 | 51.7 |
| Security Control Adjust | w/o SPA | 87.1 | 72.9 | 86.3 | 73.5 | 71.4 |
| | w/o SUM | 68.5 | 60.3 | 72.8 | 64.1 | 57.5 |
| | w/o RAG | 71.8 | 63.9 | 75.2 | 67.3 | 54.9 |
| | w/o NER | 76.3 | 65.1 | 78.4 | 68.2 | 59.3 |
| Patch Code Generation | w/o SPA | 84.6 | 62.8 | 80.4 | 68.9 | 26.8 |
| | w/o SUM | 53.5 | 45.2 | 57.9 | 48.0 | 19.1 |
| | w/o RAG | 58.0 | 48.9 | 62.3 | 52.5 | 20.7 |
| | w/o NER | 62.7 | 50.3 | 66.1 | 54.8 | 23.9 |
| Patch Tool Suggestion | w/o SPA | 94.2 | 81.0 | 91.1 | 80.3 | 66.7 |
| | w/o SUM | 71.4 | 60.8 | 72.2 | 61.9 | 48.3 |
| | w/o RAG | 77.6 | 67.3 | 79.4 | 68.1 | 52.7 |
| | w/o NER | 82.5 | 70.6 | 85.2 | 72.7 | 55.1 |
| Advisory Correlation | w/o SPA | 89.6 | 74.1 | 84.4 | 72.1 | 71.0 |
| | w/o SUM | 75.2 | 62.3 | 70.5 | 60.2 | 50.8 |
| | w/o RAG | 69.4 | 58.5 | 66.3 | 54.7 | 49.2 |
| | w/o NER | 72.1 | 60.4 | 69.7 | 58.6 | 52.1 |

## E.4. Ablation Study

To understand the contribution of each system component, we conduct an ablation study by removing different critical function modules that are mostly intensively used: NER, SPA, SUM, or RAG. Each ablation keeps all other components unchanged and the same dataset and evaluation pipeline are used.

Across all tasks, removing RAG or SUM causes the most severe performance degradation, implying their necessity for evidence grounding and context compression. SPA remains important for localizing objectives in reasoning but impacts less than retrieval-driven modules. Removing NER shows moderate performance loss, which is more pronounced in tasks requiring IOC- or actor-level normalization (e.g., Playbook Recommendation, Patch Tool Suggestion), but less in abstract reasoning tasks. These results demonstrate that named entity normalization is beneficial but not as critical as retrieval or

summarization, suggesting LLM capabilities in implicit-entity reasoning or lightweight NER alternatives.

## E.5. Token Consumption

*Table 8.* Token consumption (generation tokens) across LLMs in CYBERTEAM for different prompting strategies. Lower values indicate lower inference cost.

| Method | Cybersecurity Agent | | | Industry-Leading LLM | | | | | | | |
|---|---|---|---|---|---|---|---|---|---|---|---|
| | LY | DH | SL | GK | G5 | QW | GM | CD | L3.1 | L4 | GA |
| *Playbook Recommend* | | | | | | | | | | | |
| CoT | 1,218 | 1,561 | 1,492 | 2,387 | 2,926 | 2,658 | 2,703 | 2,515 | 3,203 | 3,371 | 1,994 |
| ToT | 2,935 | 3,412 | 3,298 | 5,721 | 6,174 | 5,932 | 6,018 | 5,712 | 6,307 | 5,189 | 3,031 |
| **Ours** | 2,181 | 2,476 | 3,205 | 6,091 | 5,568 | 5,283 | 5,301 | 6,069 | 5,582 | 4,801 | 2,298 |
| *Security Control Adjust* | | | | | | | | | | | |
| CoT | 1,271 | 1,598 | 1,532 | 2,411 | 2,968 | 2,744 | 2,811 | 2,597 | 3,293 | 3,441 | 2,954 |
| ToT | 2,901 | 3,412 | 3,289 | 5,683 | 6,107 | 5,932 | 6,028 | 5,713 | 6,321 | 6,718 | 4,012 |
| **Ours** | 2,121 | 2,443 | 2,361 | 4,016 | 5,661 | 6,248 | 5,303 | 4,072 | 6,269 | 4,892 | 4,513 |
| *Patch Code Generation* | | | | | | | | | | | |
| CoT | 2,834 | 3,231 | 2,169 | 3,671 | 4,148 | 3,882 | 3,951 | 3,772 | 4,563 | 4,832 | 4,287 |
| ToT | 5,643 | 5,384 | 5,793 | 8,347 | 9,198 | 8,673 | 8,912 | 8,573 | 9,654 | 10,276 | 9,441 |
| **Ours** | 6,607 | 6,068 | 5,971 | 8,821 | 8,412 | 7,064 | 9,189 | 9,921 | 7,642 | 7,983 | 7,316 |
| *Patch Tool Suggestion* | | | | | | | | | | | |
| CoT | 1,183 | 1,497 | 1,438 | 2,371 | 2,911 | 2,683 | 2,735 | 2,481 | 3,136 | 3,274 | 2,967 |
| ToT | 2,674 | 3,212 | 3,048 | 5,104 | 5,711 | 5,463 | 5,622 | 5,217 | 5,932 | 6,118 | 5,591 |
| **Ours** | 2,215 | 2,487 | 2,431 | 3,911 | 4,309 | 4,129 | 4,181 | 3,927 | 4,532 | 4,718 | 4,243 |
| *Advisory Correlation* | | | | | | | | | | | |
| CoT | 1,331 | 1,674 | 1,598 | 2,703 | 3,161 | 2,912 | 2,989 | 2,754 | 3,488 | 3,622 | 2,178 |
| ToT | 2,823 | 3,477 | 3,399 | 5,888 | 6,438 | 5,587 | 6,341 | 5,998 | 6,482 | 6,841 | 3,197 |
| **Ours** | 2,464 | 2,793 | 2,702 | 4,367 | 4,853 | 6,172 | 4,619 | 4,389 | 6,207 | 6,242 | 4,716 |

Overall, the token consumption analysis in Table 8 highlights several important efficiency trade-offs across prompting strategies and model families, wherein Tree-of-Thought (ToT) may incur higher token usage due to its iterative branching and self-evaluation steps, often doubling the cost of standard Chain-of-Thought (CoT). CYBERTEAM sits between CoT and ToT for most tasks, or perform colsely as ToT. Notably, CYBERTEAM typically reduces generation tokens relative to ToT while achieving comparable or improved reasoning quality (as shown in Table 3, Figure 5, etc.).

The efficiency gains of CYBERTEAM are more pronounced on smaller cybersecurity-tuned models than on frontier models (e.g., Grok, Gemini, Claude, Llama-4), suggesting that structured prompting provides greater marginal benefit for resource-constrained deployments. However, for code-intensive tasks like Patch Code Generation, all methods lead to higher token usage, reflecting the inherent complexity of structured code reasoning and the additional context required for precise patch synthesis. These results suggest that our prompting strategy provides a practical middle ground between minimal prompting (CoT) and computationally expensive deliberation (ToT), which implies scalable deployment while preserving reasoning effectiveness.

## E.6. Evaluation on non-LLM Approaches

*Table 9.* Comparison of non-LLM baselines and CYBERTEAM across multiple benchmark tasks. Non-LLM methods includes TF-IDF for Malware Identification, Regex-based IOC extractor for Signature Matching, cosine similarity for Temporal Pattern Matching, and LSTM-based text classifiers for Classification tasks. CYBERTEAM uses LLM-based reasoning with modular workflows. Best performance per task is bolded.

| Task | Non-LLM | CYBERTEAM (LLM) | CYBERTEAM (Full) |
|---|---|---|---|
| Malware Identification (F1 ↑) | 11.8 | 86.2 | **94.1** |
| Signature Matching (F1 ↑) | 54.6 | 92.3 | **99.2** |
| Temporal Pattern Matching (Sim ↑) | 52.1 | 77.8 | **93.4** |
| Attack Vector Classification (Acc ↑) | 65.2 | 90.1 | **94.5** |
| Attack Complexity Classification (Acc ↑) | 48.2 | 81.4 | **88.2** |
| Privileges Requirement Detection (Acc ↑) | 36.9 | 75.7 | **86.5** |

To provide a meaningful non-LLM comparison, we evaluate each task against a classical baseline representative of existing automation used in security operations as detailed in Table 9.

Across all evaluated tasks, CYBERTEAM consistently outperforms the non-LLM baselines, often by a substantial margin. While classical methods perform reasonably well on narrowly scoped, pattern-based tasks such as signature matching, they struggle to generalize to semantically complex or context-dependent tasks such as malware attribution and CVSS component inference. In contrast, CYBERTEAM 's modular LLM-driven workflow yields strong gains in both accuracy and robustness by leveraging contextual reasoning and task-specific prompting. These results suggest that while traditional techniques remain useful for deterministic subtasks, LLM-based workflows provide significantly greater utility in full-spectrum threat analysis and operational decision-making.

## F. Discussion

**Generalization to Unseen and Adversarial Threats.** While CYBERTEAM focuses on benchmarking LLM-assisted analysis of real-world threat-hunting data, we recognize that defenders must also handle zero-day attacks and adversarially perturbed intelligence. Our benchmark partially reflects this scenario by including emerging CVEs and attack patterns that do not appear in prompt demonstrations, enabling us to evaluate zero-shot generalization. Moreover, CYBERTEAM 's standardized operational workflow offers a generalizable process that applies to both known and novel threats. However, adversarially constructed inputs designed to systematically mislead investigation (e.g., IOC poisoning, semantic obfuscation) require a formal threat model and security evaluation framework beyond the scope of conventional benchmarking. We therefore highlight adversarial generalization as an open research challenge and an opportunity for future work at the intersection of AI security and cyber defense.

**Failure Analysis.** We conduct an additional failure-oriented analysis on two representative modules: (1) NER-based entity extraction and (2) RAG-based retrieval. In NER-based modules (e.g., Malware Identification, Infrastructure Extraction), misses often occur when threat entities appear in uncommon formats, nested constructs, or vendor-specific aliases (e.g., ransomware payload names embedded within file paths). These errors propagate downstream, degrading actor attribution and mitigation planning. RAG-based modules exhibit a different failure signature: when logs contain incomplete or ambiguous indicators, the retriever occasionally surfaces high-overlap but semantically irrelevant sources, leading to outdated or mismatched remediation suggestions despite strong generative reasoning. These observations suggest that improving domain-specific synonym handling and retrieval precision is as crucial as scaling LLM reasoning capabilities. Incorporating structured failure annotation into future iterations of CYBERTEAM will enable more robust diagnosis and principled defenses.

