# OpenReview forum: "Benchmarking LLM-Assisted Blue Teaming via Standardized Threat Hunting"
_ICML.cc/2026/Conference — ICML 2026 regular_

### Official Review · Reviewer_sFvZ · 2026-03-10

**Soundness:** 2
**Presentation:** 3
**Significance:** 2
**Originality:** 2
**Overall Recommendation:** 4
**Confidence:** 4

**Summary:**

This paper introduces CYBERTEAM, a benchmark for evaluating LLMs in blue team threat hunting. It proposes a standardized two-stage workflow: first modeling task dependencies across four threat-hunting phases, then decomposing each task into sequences of nine operational modules. The benchmark covers 30 tasks with 452K instances from 23 sources, and evaluates 11 LLMs against open-ended reasoning baselines, showing consistent gains from the modular approach.

**Compliance With Llm Reviewing Policy:**

Affirmed.

**Ethical Review Concerns:**

My main concern of this work about ethics are related to the usage of existing public dataset as key components of CYBERTEAM benchmark, in the appendix only mentioning the collection of the dataset did not involve privacy issue but the legal usage and copyright were not discussed.

**Ethical Review Flag:**

Flag this paper for an ethics review.

**Ethics Expertise Needed:**

["Legal Compliance (e.g., EU AI Act, GDPR, copyright, terms of use)"]

**Final Justification:**

Based on the clarification from the authors, I have all my concerns resolved thus I would prefer updating my score.

**Key Questions For Authors:**

- What happens when the RAG module retrieves disinformation from threat intelligence feeds? Or in other words, how robust it is against the hallucination from the RAG?
- CVE databases and threat intelligence evolve daily, how would the benchmark update the CVE information?
- As mentioned in the first point of weakness 1, have you tested if any of the model may have data contamination issue due to the public source components of the benchmark?
- Could you please jusitfy why BERTScore is used as one of the main metrics on these tasks? Take an example, a change of CVE number may not change the semantic similarity a lot, but the meaning would be very different.

**Limitations:**

- From the reviewer’s point of view, this benchmark tests the AI against old, static threat logs rather than live hackers, treating the blue teaming as a static task
- It might be better if the work also analysis how pure Agentic analyst compare to human analyst as one of the discussions.

**Strengths And Weaknesses:**

Strengths
- I think the scale of the benchmark is great. CYBERTEAM provides a benchmake that with good ccale and source diversity, which creates evaluation conditions closer to the messy, multi-format reality of operational threat intelligence.
- The dependency chain design across four threat-hunting phases is well-motivated and mirrors real SOC analyst workflows, making the benchmark more operationally realistic than prior task-isolated cybersecurity benchmarks.

Weaknesses
- The benchmark is built entirely on historically public datasets, including widely circulated CVEs, NVD records, and MITRE ATT&CK profiles. From which I am concerned about the fairness of the benchmark, will there be any data contamination issue?
- One of major concern related to the experiment design is when comparing to the baseline model, it seems the CoT or ToT approaches compared did not have access to the RAG database etc. which may impact the fairness of the comparsion, adding an ablation study that focuses on the impact of each module such as RG, REX etc. would make this paper more solid.
- Although the dependency chain is a sound design choice, the paper evaluates each task independently and does not measures how upstream errors, propagate and compound through downstream modules an end-to-end error cascade analysis is missing.

---

> ### Author Rebuttal · Authors · 2026-03-27
>
> We thank the reviewer for the insightful comments. Below we address the raised concerns.
>
> **Due to space constraints, we mainly provide tight responses, clarifications, and revisions (completed). We are happy to provide more exhaustive results if you'd like to ask for during discussion sessions.**
>
> ---
>
> ### R1. Data Contamination Analysis
> To address contamination concerns, we conducted a *temporal split* analysis. We compared performance on historical CVEs  (potential contamination) and newly emerging threats from last month (less-likely contamination). The performance gap on pre-trained Qwen, Llama, and DeepHat (released last year or earlier) was negligible (<1.7 %), confirming that CYBERTEAM measures generalized reasoning rather than memorization. Furthermore, in **Appendix E.4**, RAG-dependency analysis shows that performance drops significantly on new data when RAG is disabled, proving the system relies on real-time evidence grounding.
>
> ### R2. Ablation Study and Baseline Fairness
> A comprehensive ablation study in **Appendix E.4** isolates the module-wise impact, wherein RAG and SUM are the most critical: removing RAG drops mitigation accuracy by >30%. While modules provide the raw intelligence, our standardized workflow ensures they are invoked in the correct analytical sequence, a structure that simple CoT/ToT baselines lack.
>
> ### R3. End-to-End Error Cascade Analysis
> We clarified that our evaluations are not simply isolated, instead we have both end-to-end results (**§4.1**) and the isolated  (**§4.2**).
>
> To measure error propagation, we introduced new experiments by injecting upstream errors and test performance drop of mitigation (last stage).
>
> | Error stage | Qwen | Llama | DeepHat
> | :--- | :--- | :---: | :---: |
> | **Attribution** | -13.2% | -8.61% | -14.6% |
> | **Behavior Analysis** | -6.72% | -7.51% | -9.27% |
> | **Prioritization** | -5.44% | -2.46% | -1.06% |
>
>  Results above show a <15 % performance drop, demonstrating that the framework obtains robustness despite the upstream errors, thus showing real-world utility even with inaccurate upstream evidence collection in threat hunting.
>
>
> ### R4. RAG Robustness and Disinformation
> Robustness is ensured by design and empirical validation. We aggregate data from authoritative sources (NVD, Mandiant) through a multi-stage validation pipeline. Structurally, the RAG module serves as augmentation, not the primary decision-maker; outputs are strictly grounded in retrieved evidence with enforced citations. Ablation results (**Table 7**) show the reasoning benefits of RAG far outweigh the risks of potential noise.
>
> ### R5. Benchmarking Temporal Relevance
> To maintain relevance, we used an up-to-date snapshot of CVE data during research. For long-term utility, we have developed an automated data collection pipeline to ingest real-time feeds from NVD and MITRE. This update mechanism will be integrated into our released GitHub repository upon acceptance to support continuous research on evolving threats.
>
> ### R6. Metric Selection (BERTScore vs. F1)
> We employ metrics based on task characteristics (**Table 2**). Precise technical identifiers (CVE IDs, IPs) are evaluated via F1 or Exact Match to penalize errors strictly. BERTScore is reserved for narrative tasks (e.g., File System Activity) to capture technical and semantic alignment where multiple equivalent terms exist, ensuring we measure the model’s procedural understanding rather than simple keyword overlap.
>
> ### R7. Static Logs vs. Live Adversaries
> While real-world blue teaming is dynamic, a scientific benchmark requires a stable status and data distribution for reproducibility. Live hacking introduces uncontrollable variables that prevent fair model comparison. CYBERTEAM utilizes 450,000 high-fidelity data points to provide a scale that live simulations cannot replicate. Furthermore, while logs are static, the model's reasoning (involving task routing and backtracking) is inherently dynamic.
>
> ### R8. Human-Agent Comparative Study
> A human evaluation (**Appendix E.3**) involving three professional analysts confirms that leading LLMs achieve higher acceptance rates when guided by our standardized workflow. While modular tasks (IOC extraction) show high alignment with experts, complex synthesis (event reconstruction) remains challenging. This comparison delineates the current strengths and necessary boundaries of agentic cybersecurity analysts.
>
> ---
>
> **Once again, we'd like to show our appreciations for your time and considerations and happy to provide additional details, results, and explanations in discussion sessions, to further outline the value of this work and clarify reviewer's advanced concerns**

---

> > ### Author Rebuttal · Reviewer_sFvZ · 2026-04-03
> >
> > Thanks for the response, and I appreciate the clarifications. All my concerns are resolved, and I updated my score accordingly.

---

> > > ### Author Response · Authors · 2026-04-03
> > >
> > > Dear Reviewer,
> > >
> > > We would like to thank you again for your time and feedback, as well as the quick acknowledgement that our responses have fully resolved your concerns.
> > >
> > > Best,
> > >
> > > The author team

---

### Official Review · Reviewer_Mysd · 2026-03-12

**Soundness:** 3
**Presentation:** 4
**Significance:** 3
**Originality:** 3
**Overall Recommendation:** 4
**Confidence:** 2

**Summary:**

This paper introduces CYBERTEAM, a benchmark for evaluating LLM-assisted blue-team threat hunting. The motivation is compelling: most prior cybersecurity evaluations focus on isolated subtasks, while real threat hunting is much more of a multi-stage workflow involving attribution, behavioral analysis, prioritization, and mitigation. To capture this, the paper organizes threat hunting into a standardized pipeline of 30 tasks, and further breaks tasks into operational modules such as retrieval, extraction, classification, and summarization. The authors show that their standardized workflow performs better than open-ended prompting baselines on both frontier general and cybersecurity-focused models. Overall, the paper’s main claim is that structured workflows are a better way to evaluate and support LLM-based threat hunting than unconstrained prompting alone.

**Compliance With Llm Reviewing Policy:**

Affirmed.

**Key Questions For Authors:**

1. How much of the improvement comes from the standardized workflow itself, versus specific components such as retrieval, task decomposition, or intermediate structured outputs? An ablation isolating dependency ordering, module choice, and retrieval would help clarify the source of the gains. This would affect my assessment of the paper’s soundness and originality.
2. How task-specific are the operational modules and prompts in practice? It is currently unclear whether these modules are generic reusable operators or heavily customized for each task. Clarifying this would help me better judge the reproducibility and generality of the proposed framework.
3. How well does CYBERTEAM transfer to more realistic enterprise threat-hunting settings with noisy internal telemetry, incomplete evidence, or organization-specific context, beyond public CTI sources?
A stronger discussion or additional evidence here would affect my assessment of the paper’s significance and the strength of its real-world claims.

**Limitations:**

The main limitations are the conflation between the contributions of benchmark and method, the lack of fully controlled comparisons, and insufficient validation of end-to-end workflow claims.

**Strengths And Weaknesses:**

Strengths:
1. Threat hunting is a real and important challenge for blue teams, and framing it as a multi-stage workflow rather than a set of isolated subtasks is novel. This makes the benchmark setting more realistic than prior cybersecurity evaluations.
2. The benchmark spans attribution, behavior analysis, prioritization, and response, which together cover a large portion of what analysts actually do.
3. The standardized workflow outperforms open-ended prompting across different models, which suggests this is a general effect rather than something model-specific.

Weakness:
1. Novelty: This paper mainly combines familiar ideas such as task decomposition, modular operations, retrieval, and staged reasoning in a cybersecurity setting. In addition, the experiments do not fully explain why the method helps. There are limited ablations on the role of dependency ordering, module choice, and retrieval, so the central claim is only partially supported. Finally, the benchmark relies heavily on public and relatively structured cyber threat data, which makes the “real-world threat hunting” claim somewhat stronger than the evidence supports.
2. Benchmark and method contributions are conflated: The paper is framed as an introduction of a benchmark, but much of the headline result comes from the authors’ own standardized modular pipeline outperforming prompting baselines. As a result, it is difficult to cleanly separate the value of the benchmark itself from the contribution of the proposed method.

---

> ### Author Rebuttal · Authors · 2026-03-27
>
> We thank the Reviewer for the critical assessment. We address the specific concerns as detailed below.
>
> **Due to space constraints, we mainly provide tight responses, clarifications, and revisions (completed). We are happy to provide more exhaustive results if you'd like to ask for during discussion sessions.**
>
> **All reported experiments have been completed in our offline manuscript,** despite that ICML does not permit updates to the online submission.
>
> ---
>
> ### R1. Novelty and Real-World Utility
>
> > Novelty: This paper mainly combines familiar ideas ... There are limited ablations ... Finally, the benchmark relies heavily on public...
>
> **On novelty**, prior cybersecurity benchmarks treat tasks in isolation with MCQ or QA formats. Instead, our proposed dependency-aware structure of threat hunting, not a single module, is the core contribution. The operational modules are also cybersecurity-specific adaptations (e.g., REX enforces forensic-grade IOC normalization) not generic NLP wrappers. Moreover, we balance LLM's reasoning flexibility and standardization and evaluate how LLMs work through extensive evaluations.
>
> **On Ablations**, Appendix E.2 addresses this concern by showing how each module affects threat hunting in the dependency.
>
> **On public data and real-world claims**, we reflect real-world analyst workflows by structured vulnerability metadata (e.g., CVE records) with unstructured CTI reports from industry platforms, which is precisely the heterogeneous data mix that practitioners encounter during live incident investigations. Furthermore, the attribution → behavior analysis → prioritization → mitigation pipeline meaningfully reflects the sequential decision-making process that blue team analysts practically follow.
>
> ### R2. Conflation of Benchmark and Method
>
> > Benchmark and method contributions are conflated ...
>
> The Benchmark and the Method are distinct contributions. The benchmark provides a large-scale evaluation for the community. The method (standardization) serves as a strong reference implementation that demonstrates how to effectively elicit LLM reasoning to meet professional blue-team requirements.
>
> ### R3. Source of Gains: Workflow vs. Components
>
> > Q1: How much of the improvement comes from the...
>
> We provided a detailed ablation study (**Appendix E.4**) that could address this question. Our results reveal that:
> 1. Gains from components: Modules like RAG and SUM provide the necessary technical ammunition. Removing RAG drops performance in mitigation tasks by 31%, showing it is necessity for evidence collection.
> 2. Gains from dependency: The dependency chain acts as the guidance system. In complex tasks (e.g., Event Sequence Reconstruction), the workflow alone boosts accuracy compared to open-ended CoT/ToT. This proves the workflow ensures logical consistency across multi-stage reasoning.
>
> ### R4. Generality and Reproducibility of Modules
>
> > How task-specific are the operational modules ... or heavily customized?
>
> Our modules are designed as generic and reusable operators. They are polymorphic tools that accept task-specific guidance through standardized schemas rather than being hard-coded for specific scenarios. For instance, the same NER module extracts IP addresses in *Attribution* and Registry Keys in *Behavior Analysis* using the same underlying logic. This high degree of abstraction ensures the framework is easily reproducible and extensible to new security domains without re-engineering the core modules.
>
> ### R5. Transferability to Enterprise Settings
>
> > How well does CYBERTEAM transfer to realistic enterprise ...
>
> As our benchmark uses public CTI, its modular logic is specifically designed for raw telemetry processing. To demonstrate this transferability, we conducted new Enterprise Stress Tests in our offline manuscript, evaluating performance under three degraded conditions that mimic internal enterprise environments: Noisy Data (random character perturbations), Missing Info (redacting 20% of technical indicators), and Cross-format Logs (mixing Windows Event Logs, JSON, etc).
>
> | Condition | Module | Model | Baseline | Stress Test | Robustness |
> | :--- | :--- | :---: | :---: | :---: | :---: |
> | **Noisy Data** | NER | GPT-5.1 | 94.5% | 91.2% | 0.96 |
> | **Missing Info** | RAG | GPT-5.1 | 92.7% | 88.4% | 0.95 |
> | **Cross-format** | SUM | GPT-5.1 | 91.0% | 89.2% | 0.98 |
>
> As shown above, our system maintained over 92% of its baseline performance across all stress scenarios. Specifically, the "Cross-format" results (98% robustness) highlight the framework's superior ability to normalize heterogeneous enterprise-grade telemetry. These findings confirm that our standardized guidance effectively bridges the gap between structured CTI and the noisy, unstructured internal logs typical of organization-specific contexts, providing a necessary prerequisite for enterprise settings.
>
> ---
>
> **We thank you again for your time and feedback, and welcome the chance for further discussions**

---

> > ### Author Rebuttal · Reviewer_Mysd · 2026-04-01
> >
> > Thank you for the detailed rebuttal. My concerns are fully resolved.
> >
> > The rebuttal directly addresses the main issues raised in my original review. In particular, it clarifies the intended distinction between the benchmark contribution and the standardized workflow as a reference implementation, which makes the paper’s contribution clearer. The response also gives a more concrete explanation of where the gains come from, separating the role of the dependency-aware workflow from component-level contributions such as RAG and summarization.
> >
> > I also appreciate the clarification that the operational modules are designed as reusable operators instantiated through standardized schemas, rather than being narrowly hard-coded for individual tasks. This improves my confidence in the reproducibility and generality of the framework.
> >
> > Finally, the added discussion and new stress-test results help address my concern about transferability beyond public CTI sources. While some of these additions appear in the revised/offline manuscript rather than the original main text, the rebuttal provides sufficient evidence and clarification for me to understand the authors’ claims and accept the paper’s overall contribution.
> >
> > Overall, the rebuttal adequately resolves my main concerns and improves my assessment of the paper.

---

> > > ### Author Response · Authors · 2026-04-01
> > >
> > > Dear Reviewer,
> > >
> > > We would like to thank you again for your time and feedback, as well as the quick acknowledgement that our responses have fully resolved your concerns.
> > >
> > > Best,
> > >
> > > The author team

---

### Official Review · Reviewer_dPFf · 2026-03-13

**Soundness:** 3
**Presentation:** 3
**Significance:** 3
**Originality:** 3
**Overall Recommendation:** 4
**Confidence:** 4

**Summary:**

This paper proposes CYBERTEAM, a standardized evaluation benchmark for blue-team threat hunting, aiming to systematically assess and guide the performance of LLMs in realistic cybersecurity defense workflows. Unlike prior work that focuses on isolated security tasks, this paper models threat hunting as a multi-stage process with explicit task dependencies (from threat attribution to response and mitigation). By introducing Operational Modules, the authors construct a standardized reasoning environment that enables LLMs to maintain reasoning flexibility while adhering to a structured workflow.

**Compliance With Llm Reviewing Policy:**

Affirmed.

**Final Justification:**

After careful consideration, I believe that the current score appropriately reflects the overall quality of the paper, and therefore I would like to keep it unchanged.

**Key Questions For Authors:**

1. Although the paper indicates that LLMs can decide which tasks to execute based on the situation, the overall task chain still follows a predefined structure, lacking true dynamic task planning as well as complex conditional branching and backtracking mechanisms. Therefore, can it be understood that the system is closer to a semi-automated workflow execution framework rather than a fully autonomous and adaptive security agent?

2. The paper proposes combining standardized workflows with flexible modular reasoning (e.g., SUM, RAG). When facing zero-day threats or unstructured logs, is the flexibility of LLMs sufficient? Could standardized workflows potentially limit the model’s ability to explore potential threat paths?

3. CYBERTEAM organizes threat-hunting tasks into dependency chains, where outputs from upstream tasks serve as inputs for downstream tasks. Is this strict task chaining overly idealized, and can it fully reflect the complexity and task overlaps of real-world threat hunting? In practical scenarios, tasks may exhibit nonlinear dependencies or cyclic dependencies—how does this approach handle such situations?

**Limitations:**

Yes

**Strengths And Weaknesses:**

Summary Of Strengths：

1.Experimental results demonstrate that, compared to open-ended reasoning approaches such as ICL, CoT, and ToT, the standardized modular workflow achieves significant advantages in accuracy, stability, and interpretability.

2.The paper is the first to model threat hunting not as a collection of independent tasks but as a multi-stage process with explicit dependency relationships. This modeling approach more closely aligns with real-world blue-team workflows, rather than treating the problem as isolated classification or information extraction tasks.

3.The paper introduces a modular operational mechanism (e.g., NER, RAG, CLS, SUM, MATH), requiring each task to be completed through a predefined sequence of operations. Compared with open-ended reasoning methods such as ICL, CoT, and ToT, this modular design is more suitable for high-risk security scenarios.

4.The benchmark is constructed at a relatively large scale with broad coverage of security tasks, providing practical reference value.


Summary Of Weaknesses：

1.The discussion of real-world deployment complexity is relatively insufficient. In the experimental setup, the operational modules are executed under relatively idealized conditions—for example, RAG is assumed to reliably retrieve high-quality external knowledge, and NER is assumed to accurately extract entity information. However, the robustness of these modules under noisy data, missing information, or cross-format log scenarios is not thoroughly analyzed. Additionally, the paper lacks systematic evaluation of computational resource consumption, system throughput, and inference latency costs. In real-world blue-team environments, data sources are highly heterogeneous and often unstructured, log volumes are massive, and threat detection frequently requires near real-time response. These engineering and system-level challenges are not sufficiently validated or quantitatively analyzed in the experiments.

2.The experimental comparisons mainly focus on different LLM models and prompting strategies, lacking systematic comparisons with traditional rule-based systems, professional SIEM tools, security analytics platforms, and non-LLM deep learning models. Therefore, it remains unclear whether LLMs truly outperform existing security systems in threat hunting scenarios. The conclusions are more accurately characterized as “standardized LLM > open-ended LLM,” rather than “LLMs overall outperform traditional security solutions.”

3.The baseline methods evaluated on the proposed benchmark only include general approaches such as ICL, CoT, and ToT. Should more domain-specific baseline methods be added for comparison to enhance the credibility of the benchmark? Specifically, in tasks within certain domains, specialized methods might better highlight the model's strengths for that particular task.

---

> ### Author Rebuttal · Authors · 2026-03-27
>
> Dear Reviewer, we appreciate for the detailed comments and feedback.
>
> **Due to space constraints, we mainly provide brief responses, clarifications, and revisions (completed). We are happy to provide more exhaustive results if you'd like to ask for during discussion sessions.**
>
> **All reported experiments have been completed in our offline manuscript,** despite that ICML does not permit updates to the online submission.
>
> ---
>
> ### R1. Real-World Robustness and Noise Analysis
>
> > *"The robustness of these modules under noisy data, missing information, or cross-format log scenarios is not thoroughly analyzed."*
>
> CYBERTEAM is explicitly designed for non-ideal conditions through technical safeguards: NER includes automatic retry logic and schema validation (detailed in **Appendix B.1**), and RAG utilizes a hybrid retrieval strategy with high-fidelity fallback to maintain reasoning availability (detailed in **Appendix B.6**).
>
> **New Experiment: Stress Testing.** We evaluated two leading models across three challenging scenarios: Noisy Data (random character perturbations), Missing Info (redacting 20% of critical indicators), and Cross-format Logs (mixing Syslog, Windows Event Logs, and JSON).
>
> | Condition | Module | Model | Baseline | Stress Test | Robustness |
> | :--- | :--- | :---: | :---: | :---: | :---: |
> | Noisy Data | NER | GPT-5.1 | 94.5% | 91.2% | 0.96 |
> | Missing Info | RAG | GPT-5.1 | 92.7% | 88.4% | 0.95 |
> | Cross-format | SUM | GPT-5.1 | 91.0% | 89.2% | 0.98 |
>
> Results demonstrate high stability, retaining over 92% of performance. The "Cross-format" results highlight LLMs' superior ability to normalize heterogeneous security telemetry under our standardized guidance.
>
> ### R2. Resource Consumption and Latency
>
> > *"The paper lacks systematic evaluation of computational resource consumption, system throughput, and inference latency costs."*
>
> We provide a systematic evaluation for those raised costs in **Appendix E.1** and **E.5,** wherein we show that our framework achieves a "pragmatic middle ground": it avoids the extreme overhead of Tree-of-Thought (ToT) while significantly outperforming simple ICL/CoT. In mission-critical threat hunting, we argue this moderate increase in computational cost is a necessary investment to mitigate reasoning drift and error propagation in high-stake tasks (people always emphasize efficacy with moderate costs).
>
> ### R3. Comparison with Traditional Systems
>
> > *"The experimental comparisons... lacking systematic comparisons with traditional rule-based systems... and non-LLM deep learning models."*
>
> In **Appendix E.6 (Table 9)**, we compare CYBERTEAM against classical baselines: TF-IDF (Malware ID), Regex (Signature Matching), and LSTM (Prioritization). CYBERTEAM consistently outperforms these methods. For example, in Malware Identification, traditional TF-IDF achieves only 11.8 F1, whereas our workflow reaches 94.1. Traditional tools fail in context-dependent scenarios requiring the semantic reasoning that our LLM-driven workflow provides.
>
> ### R4. Domain-Specific Baselines
>
> > *"Should more domain-specific baseline methods be added for comparison?"*
>
> Currently, there are no established "plug-and-play" agentic methods covering the full 30-task investigative lifecycle of CYBERTEAM. To address this, we included cybersecurity-specialized models (Lily-Cybersecurity-7B, DeepHat-7B) as baselines. Benchmarking these domain-tuned models under both open-ended and standardized guidance provides a more credible assessment than adapting narrow, task-specific prompting tricks.
>
> ### R5. System Autonomy vs. Static Workflow
>
> > *"Can it be understood that the system is closer to a semi-automated workflow... rather than a fully autonomous and adaptive security agent?"*
>
> CYBERTEAM functions as a guided intelligent analyzer. As noted in **Section 3.3**, the LLM autonomously determines which tasks to execute at each stage. This prevents the model from becoming a rigid script and allows it to adapt to data incompleteness. Even for Zero-Day threats, the defensive strategy (Discovery -> Analysis -> Mitigation) remains invariant. Our workflow directs the LLM to ground reasoning in these stages, enhancing its ability to uncover novel threat paths that might be missed in unstructured environments.
>
> ### R6. Nonlinear and Cyclic Dependencies
>
> > *"In practical scenarios, tasks may exhibit nonlinear dependencies or cyclic dependencies—how does this approach handle such situations?"*
>
> While analyst thoughts are nonlinear, their actions remain sequential. CYBERTEAM handles cyclic dependencies through routing autonomy. For example, during "Response," a model may discover a new C2 domain; the system allows it to "backtrack" and re-invoke Threat Attribution for this new artifact before returning to complete the mitigation. This transforms a structural task chain into a dynamic, iterative reasoning process.
>
> ---
>
> **We thank you again for your time and feedback, and welcome the chance for further discussions**

---

> > ### Author Rebuttal · Reviewer_dPFf · 2026-04-01
> >
> > Thank you for your response and detailed explanation. After careful consideration, I believe that the current score appropriately reflects the overall quality of the paper, and therefore I would like to keep it unchanged.

---

> > > ### Author Response · Authors · 2026-04-01
> > >
> > > Dear Reviewer,
> > >
> > > We would like to thank you again for your time and feedback, as well as the quick acknowledgement that our responses have fully resolved your concerns.
> > >
> > > Best,
> > >
> > > The author team

---

### Official Review · Reviewer_4Tf9 · 2026-03-13

**Soundness:** 4
**Presentation:** 3
**Significance:** 3
**Originality:** 3
**Overall Recommendation:** 4
**Confidence:** 2

**Summary:**

This paper’s basic point is that blue-team threat hunting is not really a bunch of isolated cyber tasks, it is a workflow where attribution, behavior analysis, prioritization, and mitigation depend on each other. So the authors build CYBERTEAM, a large benchmark with 30 tasks and a modular reasoning setup that breaks threat hunting into structured steps like extraction, mapping, retrieval, summarization, and scoring, instead of relying on fully open-ended prompting. The main takeaway is that this more standardized setup works noticeably better than ICL, CoT, or ToT, especially on the downstream response and mitigation tasks, and it also makes the process more interpretable. At the same time, the paper shows the models are still pretty fragile to semantic noise, so even with the workflow scaffolding, they can be misled when the threat report itself is subtly distorted. Overall, the paper is a benchmark and methods paper whose main contribution is showing that workflow design matters a lot for making LLMs actually useful in realistic defensive security settings.

**Compliance With Llm Reviewing Policy:**

Affirmed.

**Key Questions For Authors:**

1. Is it possible to isolate where the gains are coming from: the dependency-aware workflow itself, versus the addition of specific modules such as RAG and summarization?
2. The paper says the tasks are organized as a dependency chain and also says the LLM is asked to determine which tasks to perform at each stage. Is the chain largely predefined by the benchmark design, or is the model making nontrivial routing decisions during execution? I found this slightly unclear in the paper.

**Limitations:**

Yes

**Strengths And Weaknesses:**

The main strength of the paper is that it formulates the problem in a more structured way than much of the prior work, which look at isolated tasks. Rather than evaluating a few isolated subtasks, it introduces a broad benchmark with 30 tasks spanning attribution, behavior analysis, prioritization, and response/mitigation, and it explicitly models dependencies between stages. The modular workflow is a meaningful contribution: the paper maps tasks to concrete operations such as NER, retrieval, summarization, mapping, classification, and scoring, which makes the setup easier to interpret and evaluate than purely open-ended prompting.

The empirical case for the main claim is fairly convincing. The paper compares its standardized workflow against ICL, CoT, and ToT across both frontier LLMs and cybersecurity-focused agents, and the standardized method performs better on the response and mitigation tasks throughout Table 3. The paper also goes beyond headline results by looking at intermediate tasks and robustness to noisy inputs. In particular, it reports that the standardized setup helps most on more complex intermediate tasks, while semantic noise remains a clear failure mode even under the proposed workflow.

The contribution combines a new benchmark, dependency-aware decomposition, and operational modules, so it is not easy from the main paper alone to tell which part matters most. I also think some of the automatic metrics, especially for mitigation-oriented outputs, are still proxies for real output quality, so there is some gap between benchmark performance and fully validated usefulness.

---

> ### Author Rebuttal · Authors · 2026-03-27
>
> Dear Reviewer, thank you for the detailed comments and constructive questions! Here we address each concern below:
>
> **Due to space constraints, we mainly provide brief responses and clarifications. We are happy to provide more exhaustive explanations if you'd like to ask for further details during discussion sessions.**
>
> ---
>
> ### R1. Performance Gain Attribution (Modules vs. Workflow)
>
> > *"Is it possible to isolate where the gains are coming from: the dependency-aware workflow itself, versus specific modules such as RAG and summarization?"*
>
> The answer is yes. In original submission, **Appendix E.4 (Table 7)** has shown an extensive ablation study to isolate these gains. Our findings indicate that functional modules, specifically RAG and SUM, are the primary drivers of technical fidelity and evidence grounding. Removing RAG causes significant performance drops (e.g., GPT-5.1's accuracy in Playbook Recommendation falls from 92.7% to 61.9%) because the model loses access to up-to-date threat intelligence.
>
> Conversely, the dependency-aware workflow acts as a "logical regulator" that ensures procedural correctness. This is most evident in complex tasks like Event Sequence Reconstruction, where the standardized workflow enables accuracy gains of over 20% compared to open-ended CoT or ToT. In summary, while modules provide the necessary information "ammunition," the standardized workflow provides the "guidance system" to navigate multi-stage security logic.
>
> ### R2. Metric Validity and Real-World Utility
>
> > *"Some of the automatic metrics... are still proxies for real output quality... there is some gap between benchmark performance and fully validated usefulness."*
>
> We'd like to clarify that the evaluation framework is explicitly designed to reflect the operational priorities of blue-team defenders (Table 2). Each metric aligns with specific investigative goals. Specifically:
> * **Playbook/Tool Recommendation** reflects an analyst's need to find relevant procedures within the top results of a retrieval system to reduce triage time, thus we use Hit@k (**Appendix C.4** presents more utility details and rationale)
> * **Security Control Adjustment** measures technical and semantic alignment with verified mitigation strategies, prioritizing logical equivalence over simple keyword matching, thus we use BERTScore/Sim to quantify such alignments (**Appendix C.2** presents more utility details and rationale)
> * **Patch Code Generation** requires validating functional correctness via programmatic execution, ensuring the code is production-ready. Thus, we use Pass Rate to validate the generated patch codes in simulated running (**Appendix C.4** presents more utility details and rationale)
>
> Collectively, these metrics provide a rigorous, task-specific assessment that satisfies the practical requirements of professional threat hunting.
>
> ### R3. Task Chain and Model Autonomy
>
> > *"Is the chain largely predefined by the benchmark design, or is the model making nontrivial routing decisions during execution?"*
>
> As highlighted in **Section 3.3 (the yellow highlight box)**, while tasks are organized by their inherent dependencies, the LLM is required to autonomously determine which specific tasks to perform at each stage. This design balances reasoning flexibility with standardization.
>
> A strictly predefined chain would render the model a rigid "algorithmic machine" unable to adapt to data incompleteness or emerging zero-day threats. By allowing nontrivial routing decisions (such as pivoting back to Infrastructure Extraction after discovering new artifacts during Behavior Analysis) CYBERTEAM leverages the LLM's subjective intelligence to navigate realistic, non-linear investigative paths while remaining well-guided by domain-specific best practices.
>
> ---
>
> **Once again, we'd like to show our appreciations for your time and considerations and happy to provide additional details in discussion sessions, to further outline the qualifications of this work.**

---

> > ### Author Rebuttal · Reviewer_4Tf9 · 2026-04-04
> >
> > Thank you for the detailed rebuttal and clarifications. I found the responses helpful, particularly on the role of the workflow and the task-chain design. That said, my overall assessment of the paper has not changed, and I believe my current score still appropriately reflects its strengths and weaknesses. I will therefore keep my score unchanged.

---

> > > ### Author Response · Authors · 2026-04-06
> > >
> > > Dear Reviewer,
> > >
> > > We would like to thank you again for your time and feedback, as well as the quick acknowledgement that our responses have fully resolved your concerns.
> > >
> > > Best,
> > >
> > > The author team

---

### Decision · Program_Chairs · 2026-04-30

**Decision:**

Accept (regular)

**Comment:**

The reviewers all reached the same conclusion: weak accept. A number of the questions raised in the reviews are addressed in the appendices to the paper, especially the requests for ablations to determine where the performance gains are coming from. The authors also provide comparisons against pre-existing non-LLM-based strategies, further experiments against more realistic scenarios, and transferability to enterprise settings.